# Multi-scale Consistency for Robust 3D Registration via Hierarchical Sinkhorn Tree

Chengwei Ren[1,2]    Yifan Feng[3]    Weixiang Zhang[2]    Xiao-Ping Zhang[1,2*]    Yue Gao[3*]

{[1]Shenzhen Ubiquitous Data Enabling Key Lab, [2]Shenzhen International Graduate School, [3]BNRist, THUIBCS, School of Software}, Tsinghua University

rcw22@mails.tsinghua.edu.cn, evanfeng97@gmail.com, zhang-wx22@mails.tsinghua.edu.cn
xpzhang@ieee.org, gaoyue@tsinghua.edu.cn

## Abstract

We study the problem of retrieving accurate correspondence through multi-scale consistency (MSC) for robust point cloud registration. Existing works in a coarse-to-fine manner either suffer from severe noisy correspondences caused by unreliable coarse matching or struggle to form outlier-free coarse-level correspondence sets. To tackle this, we present **H**ierarchical **S**inkhorn **T**ree (HST), a pruned tree structure designed to hierarchically measure the local consistency of each coarse correspondence across multiple feature scales, thereby filtering out the local dissimilar ones. In this way, we convert the modeling of MSC for each correspondence into a BFS traversal with pruning of a K-ary tree rooted at the superpoint, with its K nearest neighbors in the feature pyramid serving as child nodes. To achieve efficient pruning and accurate vicinity characterization, we further propose a novel overlap-aware Sinkhorn Distance, which retains only the most likely overlapping points for local measurement and next level exploration. The modeling process essentially involves traversing a pair of HSTs synchronously and aggregating the consistency measures of corresponding tree nodes. Extensive experiments demonstrate HST consistently outperforms the state-of-the-art methods on both indoor and outdoor benchmarks.

## 1 Introduction

Point cloud registration is a crucial task in 3D computer vision that involves aligning a pair of partially overlapped point clouds to create a unified scene representation. The most common approaches [1, 2, 3] follow the two stage technical roadmap, i.e., matching and transformation. They first form a set of high confident correspondences through repeatable feature descriptors, and then a robust estimator is utilized to calculate the rigid transformation. Although extensively studied over the past decades, the task remains challenging due to limited overlap, severe noise, etc.

Recent advances [4, 5, 6, 7] have made substantial progress in learning-based methods. The key idea is to train a shared network to extract point-wise features and establish reliable correspondences based on them. Inspired by the progress in image registration counterparts [8, 9, 10, 11], a coarse-to-fine strategy [12, 13] is leveraged to avoid keypoint detection and unrepeatable correspondence searching. It has demonstrated superior performance over the state-of-the-art methods. They establish sparse superpoint correspondences on the downsampled input point clouds (coarse level) and then refine them to point level to yield dense point

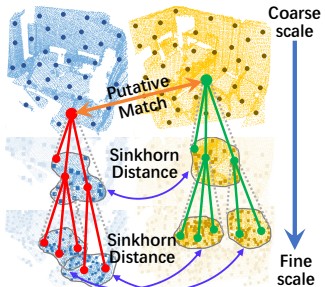

Figure 1: Illustration of the proposed HST for measuring MSC. It extracts local patches at multi-scales in the feature pyramid and then calculates their similarity layer by layer.

---

[*]Corresponding author.

38th Conference on Neural Information Processing Systems (NeurIPS 2024).

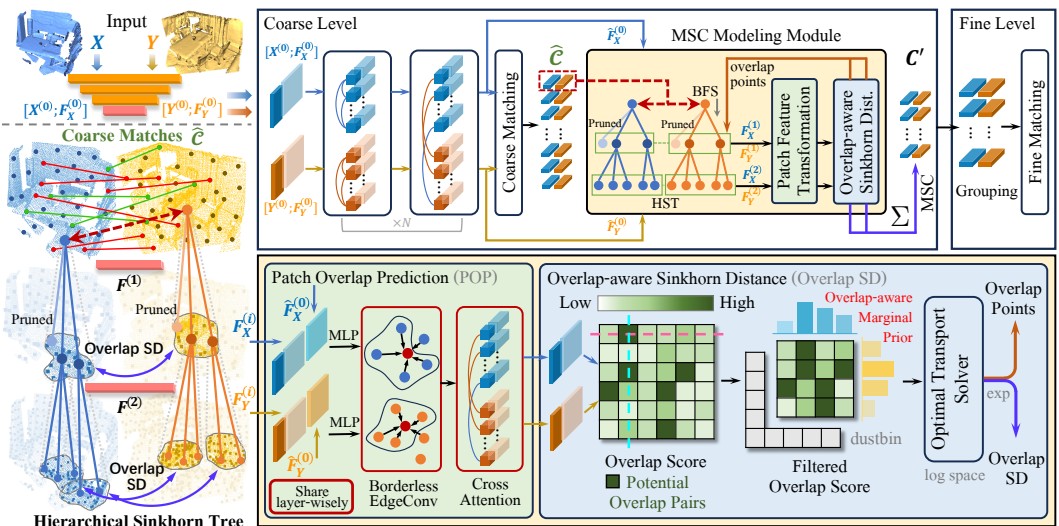

Figure 2: Overview of the proposed method. The overall architecture (**Top**) is designed in a coarse-to-fine manner, which extracts coarse correspondences and then refines them to point level. Our proposed method contains the following parts: **1.** Local exploration (**Left**) extracts local patch pairs for each putative match in multiple scales from the feature pyramid (Sec. 3.2.1). **2.** Patch Overlap Prediction module (**Bottom Middle**) is then adopted to predict the overlap points between patch pairs (Sec. 3.2.2). **3.** Overlap-aware Sinkhorn Distance (**Bottom Right**) measures the patch similarity by focusing on potential overlap points (Sec. 3.2.3). Finally, we repeat the above operations across the layers to construct the Hierarchical Sinkhorn Tree (**Left**) in a BFS way. The process of modeling MSC essentially involves traversing a pair of HSTs and then aggregating the consistency measures of the corresponding tree nodes (Sec. 3.2.4).

correspondences (fine level). Therefore, the accuracy of coarse correspondences directly impacts that of fine correspondences [12, 14]. Intuitively, A pair of incorrect coarse matches may introduce potential outlier fine correspondences, thereby influencing the final transformation estimation.

Prior works try to alleviate this problem from two perspectives. The first [13, 14, 15] attempts to nip it in the bud during coarse matching. They adapt the Transformer [16] to unordered point cloud representation to mitigate matching ambiguity. A series of embeddings are proposed to facilitate learning discriminative geometric consistency in both explicit and implicit manners for more accurate matching. The second [17, 18, 19, 20, 21, 22] ignores the inaccuracy in coarse stage and directly filters out the outlier correspondences at fine stage, known as outlier removal methods. Such methods distinguish between inlier and outlier by developing well-designed compatibilities [23, 24, 21, 22] to characterize the affinity relationship between correspondences in geometric or feature space. Though the above techniques have achieved surprising performances, both of them still suffer from noise and low overlap. Specifically, the first only considers coarse-level feature interactions, lacking exploration of finer-scale local consistency. Erroneous correspondences are more likely to occur, especially in low-overlapping scenarios. The second suffers from inaccurate outlier filtering when confronted with a set of high outlier rates correspondences caused by severe noise. Consequently, the outliers, which arise primarily from inaccurate matching in the coarse stage, would inevitably be involved in the transformation estimation step, ultimately resulting in failed registration.

To tackle the above problems, we propose the Hierarchical Sinkhorn Tree with overlap-aware Sinkhorn distance to model the multi-scale consistency (MSC) of correspondences for more accurate coarse-level matching. Multi-scale consistency analysis, due to its innate capability to analyze patterns at different levels of detail, has been variously applied to matching- [25, 26, 27] and retrieval- [28, 29] related vision tasks. However, in point cloud registration, where the irregular nature of data hinders the promotion of corresponding image MSC analysis methods, leveraging them for more reliable correspondence retrieval remains unexplored. Given that MSC characterizes the local consistency of matches across multiple feature scales, which effectively addresses the challenge of inaccurate matches at the coarse stage, we aim to tackle them both. The key idea is to hierarchically evaluate the vicinity geometric similarity of each correspondence at multiple scales using Sinkhorn distance [30] and filter out highly dissimilar coarse matches. Specifically, we employ Local Exploration to

extract local patches at each correspondence's next decoder layer. Subsequently, the overlap-aware Sinkhorn distance (overlap SD) measures the patch differences (see Fig. 3) by focusing the Sinkhorn operation on potential overlap points within the patch, disregarding the non-overlapping ones. This significantly enhances the robustness to noise points while achieving computational savings. Following this, a subset of the most likely overlapping points is retained for further local exploration and overlap SD measurement. This repetitive process refines the multi-scale local consistency of each correspondence as the scale becomes finer. Finally, we aggregate the consistency measures across all scales and utilize probabilistic search to select

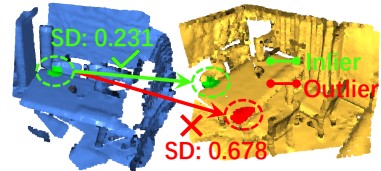

Figure 3: A toy example illustrating the Sinkhorn Distance (SD) between local patches. The patch pair from inlier correspondence (green) maintains a lower SD metric.

the most reliable coarse correspondences robustly. An overview of our method can be found in Fig. 1.

To sum up, our main contributions are three-fold:

- To the best of our knowledge, we are the first to introduce multi-scale consistency into point cloud registration task to mitigate the effects of low overlap and high noise.
- We propose a method for modeling multi-scale consistency called HST, which characterizes the similarity of potential overlapping points in the vicinity areas layer by layer and aggregates them into multi-scale consistency.
- We introduce an overlap-aware Sinkhorn Distance to focus optimal transport processes only on potential overlapping points, significantly enhancing the robustness of consistency calculations while reducing solution complexity.

Extensive experiments on both indoor and outdoor benchmarks demonstrate our scene-agnostic superiority. HST significantly outperforms the state-of-the-art methods on Registration Recall on the challenging 3DLoMatch benchmark.

## 2 Related Work

**Correspondence-based point cloud registration.** Early correspondence-based registration methods follow the detect-and-transform pipelines. A series of local geometric descriptors [1, 31, 32, 33] are proposed to detect repeatable salient points of point cloud pairs. Then point-level correspondences [5, 4, 6] are established to recover the transformation between point clouds through a robust estimator [20, 21, 22]. Recently, detector-free methods [12, 13, 15] to avoid unrepeatable keypoint detection are proposed. They introduce a coarse-to-fine manner derived from 2D image matching [8, 9, 10, 11] to shrink the correspondence searching space. It first extracts full and reliable matches on coarse-resolution features and then refines them on the corresponding local patch from finer resolution. It's becoming prevalent due to its excellent matching accuracy and computational efficiency. Our method follows the technical roadmap of detector-free methods and focuses on improving the correspondence reliability of the coarse phase.

**Coarse-to-fine matching.** Recent works [8, 9, 10, 11] advance 2D image matching by leveraging a coarse-to-fine fashion to avoid unrepeatable keypoint detection while increasing matching reliability, known as detector-free methods. DualRC-Net [8] extracts coarse feature maps to form complete correlations and generate pixel-level matches with the help of fine features. Patch2pix [9] refines coarse patch matches by regressing fine pixel matches from local regions. Loftr [10] and CasMTR [11] introduce an attention mechanism [16] to search for accurate low-resolution correspondences and then establish dense matching based on informative high-resolution local patches.

**Multi-scale consistency.** The core idea of multi-scale consistency is to evaluate the similarity of matches across multiple feature scales. The criterion under this for evaluating an inlier match is that each spatial resolution of feature pyramids should maintain a certain degree of similarity. BiseNet [34] utilizes an auxiliary loss to supervise the consistency between multi-scale features and ground-truth to enhance the feature representation. MSCAN [28] leverages a multi-scale attention module to capture features of different scales and aggregates them into a global descriptor for robust image retrieval. ACMM [25] conducts the multi-scale geometric consistency to refine depth maps at each scale, reaching satisfying performance.

## 3 Method

### 3.1 Problem Statement

Given two point clouds $\mathbf{X} = \left\{ \mathbf{x}_i \in \mathbb{R}^3 \mid i = 1..N \right\}$, and $\mathbf{Y} = \left\{ \mathbf{y}_j \in \mathbb{R}^3 \mid i = 1..M \right\}$, the goal is to recover a rigid transformation $\mathbf{T}(\mathbf{R}, \mathbf{t})$ that aligns the two point clouds with rotation $\mathbf{R} \in SO(3)$ and translation $\mathbf{t} \in \mathbf{R}^3$. We utilize the KPConv-FPN [33] as the backbone and leverage geometric self- and cross-attention [13] to estimate $C$ pairs of coarse correspondences for registration in a coarse-to-fine manner [12, 13]. Our goal is to remove the outlier correspondences from the above estimated correspondences for further accurate and robust registration. The key idea is to hierarchically model the vicinity similarity at multiple scales between each correspondence.

### 3.2 Hierarchical Sinkhorn Tree

We first propose *Multi-scale Consistency* (MSC) for coarse level correspondences, which assumes that each correspondence contains similar local features at different down-/up-sampling scales. I.e., given a pair of points $\mathbf{x}$ and $\mathbf{y}$ with their local patch features at $L$ scales to be $\{\mathbf{F_X}^{(1)}, \mathbf{F_X}^{(2)}, ..., \mathbf{F_X}^{(L)}\}$ and $\{\mathbf{F_Y}^{(1)}, \mathbf{F_Y}^{(2)}, ..., \mathbf{F_Y}^{(L)}\}$, then $\mathbf{x}$ and $\mathbf{y}$ is an putative inlier correspondence only they satisfy:

$$\sum\nolimits_{l=1}^{L} \text{dist}(\mathbf{F_X}^{(l)}, \mathbf{F_Y}^{(l)}) < \theta_d, \ l \in [1, L], \tag{1}$$

where $\text{dist}(\cdot)$ is the pre-defined metric for measuring the patch difference and $\theta_d$ is the threshold.

We further propose the *Hierarchical Sinkhorn Tree* with *Overlap-aware Sinkhorn Distance* to model the MSC of correspondences. It first applies Local Exploration (Sec. 3.2.1) to extract local patch at next finer scale from the feature pyramid. Then the Overlap-aware Sinkhorn Distance (Sec. 3.2.3) picks the overlap points from patches using Patch Overlap Prediction (Sec. 3.2.2) module and performs Sinkhorn Distance with overlap-aware marginal initialization on them. These two steps are performed layer-wisely in a hierarchical tree way called the Hierarchical Sinkhorn Tree (Sec. 3.2.4).

#### 3.2.1 Local Exploration

KPConv-FPN [33] varies point density by altering the size of grid cells at each layer to form feature pyramids. Moreover, the features from FPN decoder provide sufficient information to analyze the vicinity similarity between correspondences due to the skip connections. Consequently, we perform nearest-neighbor exploration [35] on the match's neighboring points from subsequent decoder layer.

Given point cloud $\mathbf{X}^{(l)}$ from the $l$-th layer of decoder, and its next layer point cloud $\mathbf{X}^{(l+1)}$, where $0 \leq l \leq L - 1$ and $L$ is the number of decoder. We have $k$-nearest neighbor ($k$-NN) search to obtain the local patch $\mathbf{P}_i^{(l+1)}$ from the next layer for point $\mathbf{x}_i^{(l)}$:

$$\mathbf{P}_i^{(l+1)} = \text{argtopk}_{\mathbf{x}_j^{(l+1)} \in \mathbf{X}^{(l+1)}}(-||\mathbf{x}_i^{(l)}, \mathbf{x}_j^{(l+1)}||_2) \tag{2}$$

#### 3.2.2 Overlap Points Prediction

After the above local exploration, each correspondence obtains a pair of local patch features from the first decoder layer. We then utilize them to predict overlap points between each patch pair for subsequent Overlap-aware Sinkhorn Distance Computation. We extract overlap information from local patches using a simple yet effective **Patch Overlap Prediction (POP)** module. It takes input as the coarse-grained features containing global overlap information from coarse matching module and the fine-grained features from the current decoder layer, aiming to integrate features across granularity and predict fine-grained overlap between patches. It consists of two components: a borderless EdgeConv for aggregating local features, and a cross-attention [6, 36] for interacting cross-patch information.

Though EdgeConv [37] effectively captures local information by constructing point-wise $k$-NN graphs within the patch, it restricts the search within the patch. It pushes the vicinity towards the interior of the patch when encountering boundary points, affecting the smoothness of local descriptions. Therefore, we have lifted this restriction, allowing the $k$-NN search to include points outside the patch, and we refer to it as borderless EdgeConv. Furthermore, we strengthen each point

feature as $\mathbf{f_x} = \xi_\theta(\text{cat}\left[\mathbf{f_x^d} ; \mathbf{f_x^t}\right])$ for integrating overlap features into patches, Here, $\mathbf{f_x^d}$ is the feature from the current decoder layer and $\mathbf{f_x^t}$ is the nearest upsampled feature from the last transformer block. $\xi_\theta$ is a nonlinear layer activated by Softplus and each decoder layer's point features are mapped to the same dimension, allowing for the sharing parameters of subsequent modules across scales. Then the edge feature $\mathbf{e}_{ij}$ between point pair $\mathbf{x}_i$ and $\mathbf{x}_j$ with features $\mathbf{f_x^i}$ and $\mathbf{f_x^j}$ in a patch can be calculated as $\mathbf{e}_{ij}^{(k)} = \text{h}_\theta(\text{cat}[\mathbf{f_x^i}^{(k-1)}, \mathbf{f_x^j}^{(k-1)} - \mathbf{f_x^i}^{(k-1)}])$. Here, $\text{h}_\theta$ denotes a nonlinear layer activated by LeakyReLU, $k$ denotes the $k$-th layer of EdgeConv. Finally, the point-wise feature is calculated as $\mathbf{f_x^i} = \text{h}_\theta(\text{cat}[\mathbf{f_x^i}^{(0)}, \mathbf{f_x^i}^{(1)}, \mathbf{f_x^i}^{(2)}])$, and $\mathbf{f_x^i}^{(k)}$ is max-pooled point feature from $k$-th layer.

Feature-based cross-attention is then used to achieve information interaction between two local patches corresponding to a pair of node correspondences. Given the features $\mathbf{F_X}$ and $\mathbf{F_Y}$ after EdgeConv of two patches $\mathbf{X}$ and $\mathbf{Y}$, the output feature of $\mathbf{F_X}$ after cross-attention is:

$$\mathbf{f_x^i} = \sum\nolimits_{j=1}^{|\mathbf{Y}|} a_{i,j} \left(\mathbf{f_y^j W^V}\right), \mathbf{f_x^i} \in \mathbf{F_X}, \mathbf{f_y^j} \in \mathbf{F_Y}, \tag{3}$$

where $a_{i,j} = \text{Softmax}((\mathbf{f_x^i W^Q})(\mathbf{f_y^j W^K})^\top / \sqrt{d_t})$ is the attention score between $\mathbf{F_X}$ and $\mathbf{F_Y}$.

The co-contextual feature of $\mathbf{F_Y}$ can be calculated in the same way. Now the output features $\mathbf{F^X}$ and $\mathbf{F^Y}$ contain sufficient information to achieve overlap prediction. We utilize the 0-1 scaled cosine similarity between points of two patches as the overlap score:

$$\mathbf{s} = (\frac{\mathbf{F_X} \cdot \mathbf{F_Y^\top}}{\|\mathbf{F_X}\|\|\mathbf{F_Y}\|} + 1)/2. \tag{4}$$

### 3.2.3 Overlap-ware Sinkhorn Distance

**Overlap Points Filtering**  Conducting a direct search for optimal transport across *all* points within patches may precipitate erroneous matches, consequently inducing inaccuracies in Sinkhorn distance computation. To mitigate this issue, we advocate limiting the computation of the Sinkhorn distance to the most probable overlapping points. The rationale is that the selectively filtered overlapping points exhibit a higher likelihood of successful matching, thereby mitigating the susceptibility to distance measurement errors. Therefore, we only retain points with high overlap scores for further Sinkhorn iteration to improve the robustness of optimal transport solver.

Specifically, we adopt a dynamic $k$ [38, 39] strategy to select the points with top overlap score adaptively. Intuitively, the number of potential overlap points should vary across the patches due to factors like geometry, overlap area, etc. Patch with higher overlap scores should retain more overlapping points. Therefore, we select the top-$q$ predicted scores and sum them up to represent the roughly estimated overlap points number, i.e., $k = \max\{\lfloor\sum_{i=1}^{q}\mathbf{s}_q\rfloor, 1\}$. Then points with top-$k$ overlap scores are retained as overlap points. Rows and columns devoid of overlap points are discarded, and only the remaining entries are involved in the subsequent Sinkhorn operation. This strategy effectively enhances the robustness while reducing computational overhead.

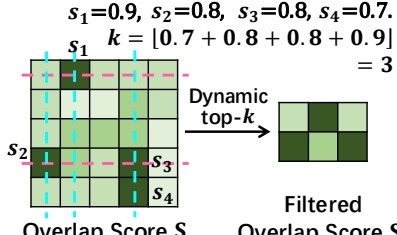

$s_1{=}0.9,\ s_2{=}0.8,\ s_3{=}0.8,\ s_4{=}0.7.$
$k = \lfloor 0.7 + 0.8 + 0.8 + 0.9 \rfloor$
$= 3$

Dynamic top-$k$

Overlap Score S        Filtered Overlap Score S'

Figure 4: Overlap Points Filtering with Dynamic Top-$k$ Illustration

**Overlap-aware Marginal Prior**  The initialization of the Sinkhorn algorithm typically sets the marginal distributions, $\mu$ and $\nu$, as uniform distributions. This initialization strategy presupposes equal importance among points to be allocated. However, it is suboptimal and potentially unreasonable in most cases. The importance of individual points ideally varies according to factors such as their positions, features, etc. With the availability of estimated overlap scores between two patches, it is prudent to incorporate this as prior during initialization. With the help of POP for estimating the overlap score, a more informed initialization of the Sinkhorn algorithm can be achieved.

Specifically, with the overlap score denoted as $\mathbf{s}$, we discard the rows and columns where without top-$k$ scores to yield the filtered overlap score $\mathbf{s}'$. Then we apply row- and column-normalization to $\mathbf{s}'$ to adjust the marginal vectors, weighting them according to the overlap scores:

$$\mu_{ov} = \sum\nolimits_{j=1}^{|\mathbf{s}_c'|} \mathbf{s}_{ij}' / \sum\nolimits_{i,j=1}^{|\mathbf{s}'|} \mathbf{s}_{ij}', \ \nu_{ov} = \sum\nolimits_{i=1}^{|\mathbf{s}_r'|} \mathbf{s}_{ij}' / \sum\nolimits_{i,j=1}^{|\mathbf{s}'|} \mathbf{s}_{ij}'. \tag{5}$$

**Overlap-aware Sinkhorn Distance**  Given a pair of patches $\mathbf{X}$ and $\mathbf{Y}$ with their overlap score $\mathbf{s}$, the filtered overlap points can be computed as $\mathbf{X_{ov}} = \left\{ \mathbf{x}_i \mid \text{top-k}_{(i,j)}(\mathbf{s}) \right\}$ and $\mathbf{Y_{ov}}$ is computed in the same way. Similar to [40], we then calculate the cost matrix $\mathbf{C} = -\mathbf{F}_{\mathbf{X_{ov}}}(\mathbf{F}_{\mathbf{Y_{ov}}})^{\top}/\sqrt{\mathbf{d}}$ based on their features. Following the dustbin setting as in [40, 41] to handle unmatched pairs, we augment the cost matrix $\mathbf{C} \in \mathbb{R}^{|\mathbf{X_{ov}}| \times |\mathbf{Y_{ov}}|}$ to $\overline{\mathbf{C}} \in \mathbb{R}^{(|\mathbf{X_{ov}}|+1) \times (|\mathbf{Y_{ov}}|+1)}$ by appending a new row and column with a learnable parameter $z$. To incorporate overlap-aware marginal prior, we append the sum of one marginal to another to serve as the dustbin. Specifically, the unnormalized $\mu_{ov}$ can be updated as $\mu_{ov} := \left[ \mu_{ov} ; \sum_{j=1}^{|\mathbf{Y_{ov}}|} \nu_{ov} \right]$, and similarly for $\nu_{ov}$. Then normalization is applied as in Eq. (5) to ensure that the sum equals 1. The formulation of the Overlap-aware Sinkhorn Distance is as follows:

$$\mathbf{D}_{\mathbf{ov}}^* = \min_{T \in U(\mu_{ov}, \nu_{ov})} \sum_{i=1}^{|\mathbf{X_{ov}}|+1} \sum_{j=1}^{|\mathbf{Y_{ov}}|+1} \mathbf{C}_{ij} \cdot T_{ij} \tag{6}$$

$$\text{s.t. } T\mathbf{1}_{|\mathbf{Y_{ov}}|+1} = \mu_{ov}, \; T^{\top}\mathbf{1}_{|\mathbf{X_{ov}}|+1} = \nu_{ov},$$

where $T_{ij}$ the $i,j$-th element from the assignment matrix $T \in R^{(|\mu_{ov}|+1) \times (|\nu_{ov}|+1)}$ of optimal transport problem. The above problem can be solved efficiently via the Sinkhorn Algorithm [42, 30] on GPUs. Moreover, it is differentiable [40], enabling the back-propagation of the supervision signal from the transport result to the Patch Overlap Prediction module.

### 3.2.4 Hierarchical Sinkhorn Tree for Multi-scale Consistency Modeling

Though the above proposed overlap SD effectively models the similarity between patches, it solely explores the neighborhood features at a single scale while still neglecting multi-scale information. To gather features at various scales for characterizing informative MSC, we propose the Hierarchical Sinkhorn Tree to traverse the feature hierarchy by repeating the above exploring and measuring steps.

Specifically, for modeling single-scale consistency, we conduct local exploration of correspondence at the next level of the feature pyramid to form local patches and then measure their similarity using overlap SD. Notably, besides the measurement computation, overlap SD also matches potential overlapping points across patches. Moreover, due to the presence of dustbin strategy, these points are further filtered. If the assignment results $T$ generated by Sinkhorn algorithm can help select more accurate overlap points for finer-grained local exploration, it enables a more precise characterization of MSC. Therefore, we retain the most likely overlapping matches via mutual top-$k$ selection (dropping dustbin). These matches continue exploring the feature at the next level with overlap SD. Repeating these two steps achieves the complete modeling of MSC across all scales. The whole modeling process essentially involves a Breath First Search (BFS) traversal with pruning of a K-ary tree rooted at the superpoint, with its $k$-NN in the feature pyramid serving as child nodes. HST is the subtrees pruned by overlap SD. Modeling MSC is essentially synchronously traversing a pair of HSTs and aggregating the overlap SD of corresponding tree nodes. We take the mean of overlap SD for each layer and then perform a weighted sum across scales to obtain the final MSC. Following [6, 14], a more robust probabilistic selection strategy is then adopted to form output correspondence set. The probability for all putative matches is $p = \text{Softmax}(1/\tau(-\mathbf{m}))$, where $\mathbf{m}$ is the 0-1 normalized MSCs and $\tau$ is the temperature parameter that controls the soft assignment.

### 3.3 Loss Functions

The total loss $L$ is the sum of each layer's loss. And each layer-wise loss $\mathcal{L}^{(l)}$ is composed of the overlap loss $\mathcal{L}_o^{(l)}$, the overlap-aware matching loss $\mathcal{L}_{om}^{(l)}$, and the Overlap-aware circle loss $\mathcal{L}_{oc}^{(l)}$, i.e., $\mathcal{L}^{(l)} = \mathcal{L}_o^{(l)} + \mathcal{L}_{om}^{(l)} + \mathcal{L}_{oc}^{(l)}$.

**Overlap loss**  To supervise the overlap prediction, we minimize the cross entropy loss between the predicted overlap score $\mathbf{s}$ and the ground truth $\overline{\mathbf{s}}$ with radius $\tau_o$. Here $[\![ \cdot ]\!]$ is the Iversion bracket.

$$\mathcal{L}_o = \frac{1}{|\mathbf{s}|} \sum_{i,j=1}^{|\mathbf{s}|} \overline{\mathbf{s}}_{ij} \log(\mathbf{s}_{ij}) + (1 - \overline{\mathbf{s}}_{ij}) \log(1 - \mathbf{s}_{ij}), \; \overline{\mathbf{s}}_{ij} = [\![ \, \|\mathbf{T}(\mathbf{x}_i) - \mathbf{y}_j\|_2 < \tau_o \, ]\!] \tag{7}$$

Table 1: Results on the 3DMatch and 3DLoMatch datasets.

| 3DMatch | | | | | | | | | | | | | | | |
|---|---|---|---|---|---|---|---|---|---|---|---|---|---|---|---|
| | RR (%) ↑ | | | | | FMR (%) ↑ | | | | | IR (%) ↑ | | | | |
| # Samples | 5000 | 2500 | 1000 | 500 | 250 | 5000 | 2500 | 1000 | 500 | 250 | 5000 | 2500 | 1000 | 500 | 250 |
| FCGF [32] | 85.1 | 84.7 | 83.3 | 81.6 | 71.4 | 97.4 | 97.3 | 97.0 | 96.7 | 96.6 | 56.8 | 54.1 | 48.7 | 42.5 | 34.1 |
| D3Feat [4] | 81.6 | 84.5 | 83.4 | 82.4 | 77.9 | 95.6 | 95.4 | 94.5 | 94.1 | 93.1 | 39.0 | 38.8 | 40.4 | 41.5 | 41.8 |
| SpinNet [5] | 88.6 | 86.6 | 85.5 | 83.5 | 70.2 | 97.6 | 97.2 | 96.8 | 95.5 | 94.3 | 47.5 | 44.7 | 39.4 | 33.9 | 27.6 |
| Predator [6] | 89.0 | 89.9 | 90.6 | 88.5 | 86.6 | 96.6 | 96.6 | 96.5 | 96.3 | 96.5 | 58.0 | 58.4 | 57.1 | 54.1 | 49.3 |
| CoFiNet [12] | 89.3 | 88.9 | 88.4 | 87.4 | 87.0 | 98.1 | 98.3 | 98.1 | 98.2 | 98.3 | 49.8 | 51.2 | 51.9 | 52.2 | 52.2 |
| YOHO [43] | 90.8 | 90.3 | 89.1 | 88.6 | 84.5 | 98.2 | 97.6 | 97.5 | 97.7 | 96.0 | 64.4 | 60.7 | 55.7 | 46.4 | 41.2 |
| GeoTR [13] | 92.0 | 91.8 | 91.8 | 91.4 | 91.2 | 97.9 | 97.9 | 97.9 | 97.9 | 97.6 | 71.9 | 75.2 | 76.0 | 82.2 | **85.1** |
| RIGA [44] | 89.3 | 88.4 | 89.1 | 89.0 | 87.7 | 97.9 | 97.8 | 97.7 | 97.7 | 97.6 | 68.4 | 69.7 | 70.6 | 70.9 | 71.0 |
| REGTR [45] | | | 92.0 | | | | | - | | | | | - | | |
| OIF [14] | 92.4 | 91.9 | 91.8 | 92.1 | 91.2 | 98.1 | 98.1 | 97.9 | 98.4 | 98.4 | 62.3 | 65.2 | 66.8 | 67.1 | 67.5 |
| RoITr [46] | 91.9 | 91.7 | 91.8 | 91.4 | 91.0 | 98.0 | 98.0 | 97.9 | 98.0 | 97.9 | **82.6** | **82.8** | **83.0** | 83.0 | 83.0 |
| HST (**Ours**) | **93.5** | **92.9** | **92.6** | 92.1 | 92.1 | **98.8** | **98.8** | **98.5** | **98.6** | **98.5** | 75.9 | 80.5 | 80.3 | **83.3** | **83.6** |

| 3DLoMatch | | | | | | | | | | | | | | | |
|---|---|---|---|---|---|---|---|---|---|---|---|---|---|---|---|
| | RR (%) ↑ | | | | | FMR (%) ↑ | | | | | IR (%) ↑ | | | | |
| # Samples | 5000 | 2500 | 1000 | 500 | 250 | 5000 | 2500 | 1000 | 500 | 250 | 5000 | 2500 | 1000 | 500 | 250 |
| FCGF [32] | 40.1 | 41.7 | 38.2 | 35.4 | 26.8 | 76.6 | 75.4 | 74.2 | 71.7 | 67.3 | 21.4 | 20.0 | 17.2 | 14.8 | 11.6 |
| D3Feat [4] | 37.2 | 42.7 | 46.9 | 43.8 | 39.1 | 67.3 | 66.7 | 67.0 | 66.7 | 66.5 | 13.2 | 13.1 | 14.0 | 14.6 | 15.0 |
| SpinNet [5] | 59.8 | 54.9 | 48.3 | 39.8 | 26.8 | 75.3 | 74.9 | 72.5 | 70.0 | 63.6 | 20.5 | 19.0 | 16.3 | 13.8 | 11.1 |
| Predator [6] | 59.8 | 61.2 | 62.4 | 60.8 | 58.1 | 78.6 | 77.4 | 76.3 | 75.7 | 75.3 | 26.7 | 28.1 | 28.3 | 27.5 | 25.8 |
| CoFiNet [12] | 67.5 | 66.2 | 64.2 | 63.1 | 61.0 | 83.1 | 83.5 | 83.3 | 83.1 | 82.6 | 24.4 | 25.9 | 26.7 | 26.8 | 26.9 |
| YOHO [43] | 65.2 | 65.5 | 63.2 | 56.5 | 48.0 | 79.4 | 78.1 | 76.3 | 73.8 | 69.1 | 25.9 | 23.3 | 22.6 | 18.2 | 15.0 |
| GeoTR [13] | 75.0 | 74.8 | 74.2 | 74.1 | 73.5 | 88.3 | 88.6 | 88.8 | 88.6 | 88.3 | 43.5 | 45.3 | 46.2 | 52.9 | 57.7 |
| RIGA [44] | 65.1 | 64.7 | 64.5 | 64.1 | 64.8 | 85.1 | 85.0 | 85.1 | 84.3 | 85.1 | 32.1 | 33.4 | 34.3 | 34.5 | 34.6 |
| REGTR [45] | | | 64.8 | | | | | - | | | | | - | | |
| OIF [14] | 76.1 | 75.4 | 75.1 | 74.4 | 73.6 | 84.6 | 85.2 | 85.5 | 86.6 | 87.0 | 27.5 | 30.0 | 31.2 | 32.6 | 33.1 |
| RoITr [46] | 74.7 | 74.8 | 74.8 | 74.2 | 73.6 | **89.6** | **89.6** | **89.5** | **89.4** | **89.3** | **54.3** | **54.6** | **55.1** | 55.2 | 55.3 |
| HST (**Ours**) | **77.8** | **77.4** | **76.9** | **75.5** | **74.0** | 88.8 | 88.8 | 89.0 | 88.8 | 89.0 | 41.7 | 48.0 | 54.5 | **56.6** | **58.3** |

**Overlap matching loss**  To supervise the optimal transport result of Overlap-aware Sinkhorn Distance, we minimize the negative log-likelihood loss as in [40] on the assignment matrix $T$. Given the set $\mathcal{OV}$ representing the overlap point pairs within the overlap radius $\tau_{ov}$ as the ground truth matching, $\mathcal{I}$ and $\mathcal{J}$ denoting the unmatched points, $\mathcal{L}_{om}$ is defined as:

$$\mathcal{L}_{om} = - \sum_{(x,y)\in\mathcal{OV}} \log T_{x,y} - \sum_{x\in\mathcal{I}} \log T_{x,m+1} - \sum_{y\in\mathcal{J}} \log T_{n+1,y}. \qquad (8)$$

**Overlap-aware circle loss**  Inspired by [34], we extend the coarse matching loss [13] to the feature hierarchy to further supervise the overlap prediction at each layer, i.e., $\mathcal{L}_{oc} = \left( \mathcal{L}_{oc}^{\mathbf{X}} + \mathcal{L}_{oc}^{\mathbf{Y}} \right)/2$, and

$$\mathcal{L}_{oc}^{\mathbf{X}} = \frac{1}{|\mathcal{A}|} \sum_{i=1}^{|\mathcal{A}|} \log[1 + \sum_{\mathcal{G}_j^{\mathbf{Y}}\in\varepsilon_p^i} e^{\lambda_i^j \beta_p^{i,j}\left(d_i^j - \Delta_p\right)} \sum_{\mathbf{G}_k^{\mathbf{Y}}\in\varepsilon_n^i} e^{\beta_n^{i,k}\left(\Delta_n - d_i^k\right)}]. \qquad (9)$$

We provide details on the individual terms and further particulars in the supplementary material.

# 4   Experiments

We evaluate our proposed Hierarchical Sinkhorn Tree on both indoor 3DMatch [1], 3DLoMatch [6] dataset, and outdoor KITTI odometry [47] dataset. More details about the datasets, evaluation metrics, and implementation are provided in the supplementary material.

## 4.1   Indoor 3DMatch & 3DLoMatch

**Metrics.**  We adopt 5 metrics to evaluate our method. Our main metric is the Registration Recall (RR), the fraction of correctly aligned point cloud pairs. Following the settings in [6, 13], the

Table 2: Registration results w/ and w/o RANSAC on 3DMatch and 3DLoMatch. The number of samples for RANSAC-50k, weighted SVD, and LGR are 5000, 250, and all respectively. The units for metrics are RR (%), RRE (°), RTE (m), and time (s).

| Model | Estimator | 3DMatch | | | 3DLoMatch | | | Time(s) |
|---|---|---|---|---|---|---|---|---|
| | | RR | RRE | RTE | RR | RRE | RTE | |
| Predator [6] | RANSAC-50k | 89.0 | 2.02 | 0.064 | 62.5 | 3.04 | 0.093 | 3.915 |
| CoFiNet [12] | RANSAC-50k | 89.3 | 2.44 | 0.067 | 67.5 | 5.44 | 0.155 | **1.746** |
| GeoTrans [13] | RANSAC-50k | 92.0 | 1.87 | 0.065 | 75.0 | 2.94 | 0.090 | 1.992 |
| OIF-PCR [14] | RANSAC-50k | 92.4 | 1.86 | 0.064 | 76.1 | 3.04 | 0.092 | - |
| RoReg [48] | RANSAC-50k | 92.9 | 1.84 | **0.063** | 70.3 | 3.09 | 0.093 | - |
| RoITr [46] | RANSAC-50k | 92.4 | 1.87 | **0.063** | 75.7 | 2.84 | 0.091 | 2.270 |
| HST (**Ours**) | RANSAC-50k | **93.5** | **1.83** | **0.063** | **77.8** | **2.80** | **0.088** | 2.160 |
| Predator [6] | weighted SVD | 50.0 | 3.89 | 0.122 | 6.4 | >10 | >1 | **0.073** |
| CoFiNet [12] | weighted SVD | 64.6 | 2.92 | 0.087 | 21.6 | 6.34 | 0.154 | 0.264 |
| GeoTrans [13] | weighted SVD | 86.5 | 2.13 | 0.070 | 59.9 | 3.88 | **0.105** | 0.245 |
| HST (**Ours**) | weighted SVD | **88.1** | **2.10** | **0.068** | **62.1** | **3.78** | **0.105** | 0.279 |
| CoFiNet [12] | LGR | 87.6 | 2.23 | 0.071 | 64.8 | 3.49 | 0.124 | 0.279 |
| GeoTrans [13] | LGR | 91.5 | 1.91 | 0.068 | 74.0 | 2.95 | 0.090 | **0.260** |
| RoItr [46] | LGR | 91.5 | 1.80 | 0.065 | 73.7 | 2.88 | 0.090 | 0.311 |
| HST (**Ours**) | LGR | **93.2** | **1.70** | **0.059** | **77.3** | **2.71** | **0.084** | 0.296 |
| PEAL-3d [15] | Iterative LGR | 94.1 | 1.75 | 0.061 | 78.8 | 2.80 | 0.087 | - |
| HST (**Ours**) | Iterative LGR | **94.4** | **1.72** | **0.061** | **79.9** | **2.72** | **0.084** | - |

registration is considered correct if the root mean square error (RMSE) is under 0.2m. We also report the feature matching recall (FMR) and inlier ratio (IR). IR is defined as the fraction of point pairs' distance less than 0.1m under correct transformation and FMR is the fraction of point pairs' IR larger than 5%. Relative Rotation Error (RRE) and Relative Translation Error (RTE) are the metrics to measure the difference between the predicted transformation and the ground-truth transformation.

**Comparisons to the state-of-the-art methods.** We compare our proposed HST to recent state-of-the-art methods including: FCGF [32], D3Feat [4], SpinNet [5], Predator [6], CoFiNet [12], YOHO [43], GeoTransformer (GeoTR) [13], RIGA [44], REGTR [45], OIF-PCR (OIF) [14] and RoITr [46], see in Tab. 1. We first report the RR, FMR, and IR results using the RANSAC [17] estimator under different numbers of sampled correspondences. Following [6, 13, 14], we run RANSAC for 50k iterations to estimate the final transformation. Following [14], we only report individual results of REGTR [45] not distinguishing the number of sampling points because it estimates the final pose based on all the superpoints instead of sampling. HST outperforms all the previous descriptor-based and end-to-end methods on both 3DMatch and 3DLoMatch. It surpasses the closest competitor by 1.1 pp and 1.7 pp respectively on RR, reflecting the superiority in actual high and low overlap alignment. When the correspondence number varies, HST consistently maintains a significant lead over the others on the boards. For FMR and IR, HST still shows outstanding performance. It reaches the best or second best in most data points, indicating its stability when confronting limited correspondences.

We further compare the registration results replacing RANSAC with other estimators including weighted SVD [49], local-to-global registration (LGR) [13], and Iterative LGR [15] in Tab. 2. In addition to RR, we further introduce RRE (°), RTE (m), and time (s) to evaluate the estimated error and latency of the methods. First, when replacing with weighted SVD, all methods suffer severe performance degradation and some even fail to align while HST still achieved the best performance across all the metrics. When using LGR, HST consistently outperforms all the state-of-the-art methods by a large margin. It improves the previous best ([13]) by 1.7% on 3DMatch [1] and 3.6% on 3DLoMatch [6] while with the smallest RRE and RTE. The most recent advances [15] refine the registration in an iterative update way. We adapt HST to iterative registration built upon PEAL [15] with 3d overlap prior and utilize LGR multiple times to gradually refine the final transformation. The results demonstrate that HST achieving competitive performance although we have not optimized it for the multi-step pipeline, which once again proves the effectiveness of our method. The above results demonstrate that our method performs well even without RANSAC to filter out outliers in

Table 3: Comparisons to the baseline w/o and w/ fine-tuning on 3DExMatch using different estimators.

| Method | Estimator | # samples | w/o tuning | | | w/ tuning | | |
|--------|-----------|-----------|-----------|-----------|-----------|-----------|-----------|-----------|
| | | | RR(%) | RRE(°) | RTE(cm) | RR(%) | RRE(°) | RTE(cm) |
| GeoTrans | LGR | - | 29.8 | 3.83 | **0.110** | 53.8 | **3.73** | 0.107 |
| HST (**Ours**) | LGR | - | **32.5** | **3.78** | 0.111 | **59.0** | 3.79 | **0.106** |
| GeoTrans | RANSAC | 250 | 28.8 | 4.56 | 0.123 | 47.0 | **4.39** | 0.121 |
| HST (**Ours**) | RANSAC | 250 | **31.5** | **4.42** | **0.122** | **49.8** | 4.43 | **0.119** |
| GeoTrans | RANSAC | 500 | 30.5 | 4.25 | 0.121 | 49.9 | 4.55 | **0.113** |
| HST (**Ours**) | RANSAC | 500 | **32.5** | **4.16** | **0.118** | **52.6** | **4.03** | 0.117 |
| GeoTrans | RANSAC | 1000 | 31.7 | 4.23 | 0.118 | 52.4 | **4.03** | 0.117 |
| HST (**Ours**) | RANSAC | 1000 | **33.6** | **4.03** | **0.114** | **57.5** | 4.13 | **0.115** |
| GeoTrans | RANSAC | 2500 | 31.9 | 4.09 | 0.116 | 55.1 | **3.89** | 0.116 |
| HST (**Ours**) | RANSAC | 2500 | **34.7** | **3.81** | **0.111** | **59.0** | **3.89** | **0.112** |
| GeoTrans | RANSAC | 5000 | 31.0 | 4.04 | 0.117 | 56.1 | 4.07 | 0.116 |
| HST (**Ours**) | RANSAC | 5000 | **34.4** | **3.87** | **0.114** | **60.3** | **3.93** | **0.111** |

fine-level correspondences. It indicates that HST is capable of forming reliable and robust coarse matching sets, which effectively assist subsequent fine matching to achieve accurate registration.

**Comparisons under extreme low overlap.** We design a set of experiments targeting overlap ratios of less than 10% and compare the performance with GeoTransformer [13] to test the robustness of HST when facing extremely low overlap. However, the currently available preprocessed datasets, 3DMatch [1] and 3DLoMatch [6] ( > 30% and 10 - 30%, respectively), do not include samples with such extremely low overlap (< 10%). Therefore, we access the 3DMatch raw dataset [1] and collect a new set of point cloud pairs with overlap ratio under 10%, which we refer to as 3DExtremeLoMatch (3DExMatch) dataset, containing a total of 1,343 samples. We first test the model pre-trained on 3DMatch directly on 3DExMatch, and the results can be found in the left part of Tab. 3.

We then randomly divide the 3DExMatch into training, validation, and test sets with proportions of 60% (805 samples), 10% (134 samples), and 30% (404 samples), respectively. We fine-tune the pretrained GeoTransformer [13] and HST both for 3 epochs on the training set, and then evaluate them on the test set. The results can be found in right part of Tab. 3. Both results clearly demonstrate that HST maintains strong performance even under low overlap conditions, confirming the effectiveness of our method.

**Comparisons to the outlier-rejection methods.** We further compare HST with other state-of-the-art outlier-rejection methods to gain more insights into handling outliers. For fairness, we replace HST directly with GC-RANSAC [18], MAC [22], and FastMAC [50]. Results on both 3DMatch and 3DLoMatch can be found in the following Tab. 4. All three methods, along with HST, showed performance improvements compared to the vanilla GeoTransformer, with HST demonstrating the most significant enhancement. However, on 3DLoMatch, the improvements from these three methods were less pronounced, and some even showed a potential negative impact. In contrast, HST maintained better robustness, highlighting its superior effectiveness over previous outlier rejection methods in scenarios with low overlap and high noise.

Table 4: Registration results compare with state-of-the-art outlier-rejection methods.

| Benchmark | 3DMatch | | | 3DLoMatch | | |
|-----------|---------|-----------|-----------|-----------|-----------|-----------|
| | RR(%) | RRE(°) | RTE(cm) | RR(%) | RRE(°) | RTE(cm) |
| vanilla [13] | 91.5 | 1.91 | 0.068 | 74.0 | 2.95 | 0.090 |
| GC-RANSAC [18] | 92.1 | 1.78 | 0.068 | 73.4 | 2.96 | 0.088 |
| MAC [22] | 92.2 | 1.99 | 0.067 | 74.4 | 2.85 | 0.086 |
| FastMAC [50] | 91.9 | 1.73 | 0.062 | 74.2 | 2.86 | 0.087 |
| HST (**Ours**) | **93.2** | **1.70** | **0.059** | **77.3** | **2.71** | **0.084** |

Table 6: Ablation study for each component. Tested with RANSAC # Samples=5000.

| Model | 3DMatch | | | 3DLoMatch | | | Time(s) |
| --- | --- | --- | --- | --- | --- | --- | --- |
| | IR(%) | FMR(%) | RR(%) | IR(%) | FMR(%) | RR(%) | |
| **Full (# Depth=2, all scales)** | **75.9** | **98.8** | **93.5** | **41.7** | **88.8** | **77.8** | **2.160** |
| vanilla SD | 66.0 | 98.1 | 90.4 | 37.9 | 86.4 | 73.3 | - |
| w/o Overlap Filtering | 70.5 | 98.1 | 91.6 | 39.7 | 86.5 | 75.1 | - |
| w/o Overlap-aware Initial | 71.1 | 98.6 | 92.9 | 39.3 | 88.4 | 77.0 | - |
| w/o Patch Overlap Pred | 69.8 | 98.2 | 92.0 | 39.4 | 86.8 | 75.6 | - |
| HST # Depth=1 | 74.3 | 98.2 | 92.8 | 40.5 | 88.1 | 76.8 | 2.075 |
| HST # Depth=0 | 71.0 | 98.1 | 92.4 | 39.9 | 87.5 | 74.6 | 2.039 |

## 4.2 Outdoor KITTI Odometry

**Metrics.** Following [6, 13], 3 metrics are adopted to evaluate our methods: Relative Rotation Error (RRE), Relative Translation Error (RTE), and Registration Recall (RR).

**Registration Results.** We compare our method with recent state-of-the-art methods in Tab. 5, including FCGF [32], D3Feat [4], SpinNet [5], Predator [6], CoFiNet [12], GeoTR [13], OIF [14], and PEAL [15]. Our method outperforms all the baselines for all metrics. It indicates that HST is effective in handling both indoor and outdoor scenes.

Table 5: Registration results on KITTI odometry.

| Method | RRE(°)↓ | RTE(cm)↓ | RR(%)↑ |
| --- | --- | --- | --- |
| FCGF [32] | 0.30 | 9.5 | 96.6 |
| D3Feat [4] | 0.30 | 7.2 | **99.8** |
| SpinNet [5] | 0.47 | 9.9 | 99.1 |
| Predator [6] | 0.28 | 6.8 | **99.8** |
| CoFiNet [12] | 0.41 | 8.2 | **99.8** |
| GeoTR [13] | 0.24 | 6.8 | **99.8** |
| OIF [14] | **0.23** | 6.5 | **99.8** |
| PEAL [15] | **0.23** | 6.8 | **99.8** |
| HST (**Ours**) | **0.23** | 6.3 | **99.8** |

## 4.3 Ablation Study

**Importance of individual modules.** Tab. 6 shows the results of the ablation studies of each component on 3DMatch and 3DLoMatch. We first replace our proposed overlap-aware Sinkhorn Distance with vanilla Sinkhorn Distance, i.e., removing overlap filtering and overlap-aware initialization. Results indicate that all the metrics decrease significantly, proving that HST benefits from our designed scheme. Then we ablate these two components and the POP module individually. Removing either will cause severe performance degradation, indicating each component is beneficial for patch difference measurement. Finally, we ablate the depth of HST, i.e., the number of scales used. The results show that performance degrades as the depth reduces, with optimal performance achieved when all scales are explored. This confirms that robust coarse matching relies on accurate MSC.

**Robustness study** Fig. 5 shows the results of adding zero-mean Gaussian noise with a standard deviation of 0.01 to 3DLoMatch [6] dataset, and gradually increasing the proportion of noise points to test the robustness of the model. It demonstrates that HST maintains the most stable performance compared to [13, 12], indicating its superior noise resistance.

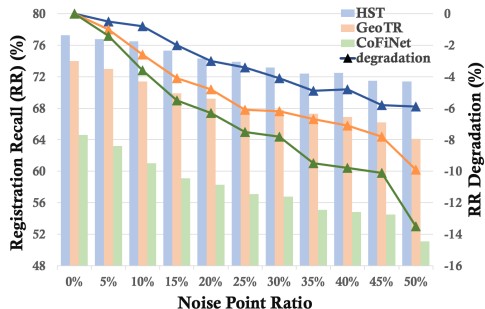

Figure 5: Details of robustness study.

## 5 Conclusion

In the paper, we present a simple but effective Hierarchical Sinkhorn Tree (HST) to model the multiscale geometric consistency for robust point cloud registration. We hierarchically explore the neighborhoods of each correspondence in their feature pyramids, and devise a novel overlap-aware Sinkhorn Distance to compute the vicinity similarity. Subsequently, the most likely overlapping points are retained to continue local exploration. The modeling process essentially involves a BFS traversal of a k-ary tree rooted at the coarse-level point. Pruning is performed during traversal using the overlap-aware Sinkhorn distance to obtain subtrees, which is the so-called HST. Extensive experiments show HST consistently outperforms the state-of-the-art methods on both indoor and outdoor benchmarks.

## Acknowledgments and Disclosure of Funding

This work was supported in part by Shenzhen Key Laboratory of Ubiquitous Data Enabling (No. ZDSYS20220527171406015), and by Tsinghua Shenzhen International Graduate School-Shenzhen Pengrui Endowed Professorship Scheme of Shenzhen Pengrui Foundation.

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

# A    Appendix / supplemental material

In this supplemental material, we first provide detailed introductions about our utilized datasets in Sec. A.1. Metrics utilized for evaluation in our experiments are demonstrated in Sec. A.2. Detailed loss and further particulars are provided in Sec. A.3. We further provide more implementation details in Sec. A.4. More results are provided in Sec. A.5. Limitations and broader impact are discussed in Sec. A.6 and Sec. A.7. Finally, we demonstrate more qualitative results of registration on 3DMatch, 3DLoMatch in Sec. A.8.

## A.1    Datasets

**3DMatch & 3DLoMatch**    3DMatch [1] contains 62 scenes collected from SUN3D [2], 7-Scenes [3], RGBD Scenes v.2 [4], Analysis-by-Synthesis [5], BundleFusion [6], and Halbel et al. [7] among which 46 scenes are used for training, 8 scenes for validation and 8 scenes for testing. Scenes are captured from various indoor environments with sensors like Microsoft Kinect. The licenses for each component are listed as follows in Tab. 7. Following [8, 9, 10], we use the training data preprocessed by [8] and evaluate on 3DMatch and 3DLoMatch benchmarks. The original 3DMatch only contains point clouds with overlap > 30%. 3DLoMatch additionally includes a subset of point cloud pairs with an overlap ratio between 10% and 30%, specifically designed to evaluate performance in low overlap scenarios.

Table 7: Datasets in 3DMatch [1] and their licenses.

| Datasets | License |
|---|---|
| SUN3D [24] | CC BY-NC-SA 4.0 |
| 7-Scenes [17] | Non-commercial use only |
| RGB-D Scenes v.2 [11] | (License not stated) |
| Analysis-by-Synthesis [21] | CC BY-NC-SA 4.0 |
| BundleFusion [5] | CC BY-NC-SA 4.0 |
| Halbel et al. [8] | CC BY-NC-SA 4.0 |

**KITTI Odometry**    KITTI [11] is a vision benchmark especially focusing on outdoor scenes. It consists of 11 sequences of outdoor driving scenarios scanned by a Velodyne HDL-64 3D laser scanner. We follow [12, 13, 8] and adopt 0-5 for training, 6-7 for validation and 8-10 for testing. The ground truth transformations are refined utilizing ICP following [12, 13, 8] due to the noisy GPS data.  we only use point cloud pairs that are at least 10m away for evaluation, i.e., 1358 pairs for training, 180 for validation, and 555 for testing following [12]. It is published under the NonCommercial-ShareAlike 3.0 License.

## A.2    Evaluation Metrics

**Relative Rotation Error**    Relative Rotation Error (RRE) is the geodesic distance in degrees between estimated and ground-truth rotation matrices. It is calculated by measuring the difference between the estimated and true rotation vectors as follows:

$$\text{RRE} = \arccos\left(\frac{\text{trace}\left(\mathbf{R}^T \cdot \overline{\mathbf{R}} - 1\right)}{2}\right). \tag{10}$$

**Relative Translation Error**    Relative Translation Error (RTE) is the Euclidean distance between estimated and ground-truth translation vectors. It is calculated by measuring the difference between the estimated and true translation matrices as follows:

$$\text{RTE} = \|\mathbf{t} - \overline{\mathbf{t}}\|_2. \tag{11}$$

**Registration Recall**    Registration Recall (RR) is the fraction of correctly registered point cloud pairs. In the 3DMatch and 3DLoMatch benchmarks, two fragments are considered registered correctly if

the transformation Root Mean Squared Error (RMSE) is smaller than 0.2m:

$$\text{RMSE} = \sqrt{\frac{1}{|\mathcal{C}^*|} \sum_{\left(\mathbf{p}^*_{x_i}, \mathbf{q}^*_{y_i}\right) \in \mathcal{C}^*} \left\| \mathbf{T}\left(\mathbf{p}^*_{x_i}\right) - \mathbf{q}^*_{y_i} \right\|^2_2}, \tag{12}$$

$$\text{RR} = \frac{1}{M} \sum_{i=1}^{M} [\![ \text{RMSE}_i < 0.2\ \text{m}]\!], \tag{13}$$

where $\mathbf{T}$ is the predicted transformation and $\mathcal{C}^*$ are the ground truth correspondences. $[\![\cdot]\!]$ is the Iversion bracket. Moreover, we align with the evaluation protocol in 3DMatch [1], the immediately adjacent point clouds are excluded due to their high overlap ratio.

In the KITTI Odometry benchmark, the registration is correct when RRE < 5° and RTE < 2m:

$$\text{RR} = \frac{1}{M} \sum_{i=1}^{M} [\![ \text{RRE}_i < 5° \wedge \text{RTE}_i < 2m ]\!]. \tag{14}$$

**Inlier Ratio**   Inlier Ratio (IR) is the fraction of inlier correspondences among all the putative correspondences. A correspondence is considered an inlier when the distance between two points is smaller than $\tau_1 = 10\ \text{cm}$ under the ground-truth transformation $\overline{\mathbf{T}}$:

$$\text{IR} = \frac{1}{|\mathcal{C}|} \sum_{\left(\mathbf{p}_{x_i}, \mathbf{q}_{y_i}\right) \in \mathcal{C}} [\![ \left\| \overline{\mathbf{T}}\left(\mathbf{p}_{x_i}\right) - \mathbf{q}_{y_i} \right\|_2 < \tau_1 ]\!], \tag{15}$$

where $\mathcal{C}$ is the putative correspondence set.

**Feature Matching Recall**   Feature Matching Recall (FMR) is the fraction of point cloud pairs whose IR is larger than a certain threshold $\tau_2 = 0.05$. It measures the likelihood that the optimal transformation between two point clouds can be recovered using robust estimators. It is calculated as:

$$\text{FMR} = \frac{1}{M} \sum_{i=1}^{M} [\![ \text{IR}_i > \tau_2 ]\!]. \tag{16}$$

### A.3   Detailed Loss

**Overlap-aware circle loss**   Each layer-wise overlap-aware circle loss [10] is calculated as the mean of losses on point cloud pairs $\mathbf{X}$ and $\mathbf{Y}$ from the $l$-th layer of feature pyramid, i.e., $\mathcal{L}_{oc}{}^{(l)} = \left( \mathcal{L}_{oc}^{\mathbf{X}(l)} + \mathcal{L}_{oc}^{\mathbf{Y}(l)} \right)/2$. The anchor patch set $\mathcal{A}$ is collected as the patches in $\mathbf{X}$ have at least one positive patch in $\mathbf{Y}$. The positive patch set $\varepsilon_p^i$ is defined as the patches that share at least 10% overlap. The negative patch set $\varepsilon_n^i$ is the patches share no overlap. Then the individual loss for $\mathbf{X}$ is defined as:

$$\mathcal{L}_{oc}^{\mathbf{X}(l)} = \frac{1}{|\mathcal{A}|} \sum_{i=1}^{|\mathcal{A}|} \log[1 + \sum_{\mathcal{G}_j^{\mathbf{Y}} \in \varepsilon_p^i} e^{\lambda_i^j \beta_p^{i,j}\left(d_i^j - \Delta_p\right)} \sum_{\mathcal{G}_k^{\mathbf{Y}} \in \varepsilon_n^i} e^{\beta_n^{i,k}\left(\Delta_n - d_i^k\right)}], \tag{17}$$

where $d_i^j = \|\hat{\mathbf{h}}_i^{\mathbf{X}} - \hat{\mathbf{h}}_j^{\mathbf{Y}}\|_2$ is the distance in the feature space, $\lambda_i^j = \left(o_i^j\right)^{\frac{1}{2}}$ and $o_i^j$ are the overlap ratio between $\mathcal{G}_i^{\mathbf{X}}$ and $\mathcal{G}_j^{\mathbf{Y}}$. The positive and negative weights are computed as $\beta_p^{i,j} = \gamma\left(d_i^j - \Delta_p\right)$ and $\beta_n^{i,k} = \gamma\left(\Delta_n - d_i^k\right)$. The hyper-parameters for margin are set to $\Delta_p = 0.1$ and $\Delta_n = 1.4$ following [10]. The loss $\mathcal{L}_{oc}^{\mathbf{Y}(l)}$ for $\mathbf{Y}$ are calculated in the same way.

Inspired by [14] introducing segmentation loss to multiple feature scales to enhance the feature representation, we extend the coarse (superpoint) matching loss to the feature hierarchy to further

supervise the overlap prediction at each layer. The overall overlap-ware circle loss is calculated as $\mathcal{L}_{oc} = \sum_{l=0}^{L} w_l \, \mathcal{L}_{oc}^{(l)}$, where $w_l$ is the weight for the $l$-th layer. For stable training and better performance, we decay the weight by half with each successive layer, i.e., $\mathcal{L}_{oc} = \sum_{l=0}^{L} (\frac{1}{2})^l \mathcal{L}_{oc}^{(l)}$.

## A.4 Implementation Details

Our proposed method is implemented and evaluated in Pytorch [15] and we train it on a single RTX 3090 GPU with an AMD EPYC 9654 CPU. Specifically, the network is trained with Adam optimizer [16] for 40 epochs on 3DMatch and 80 epochs on KITTI with batch size of 1 and weight decay of $10^{-6}$. The learning rate initializes from $10^{-4}$ and decays exponentially by 0.05 every epoch on 3DMatch and every 4 epochs on KITTI, respectively. The matching radius $\tau$ is set as 5cm for indoor 3DMatch/3DLoMatch and 60cm for outdoor KITTI to generate overlapping matches. We randomly sample 128 pairs of ground-truth coarse matches during training. The overall Multi-scale Consistency (MSC) is calculated as the weighted sum of each layer's mean overlap-aware Sinkhorn distance (overlap SD) $\mathcal{S}^{(l)}$ and the weights also decay by half as the layer increases, i.e., $\text{MSC} = \sum_{l=1}^{L}(\frac{1}{2})^l \mathcal{S}^{(l)}$. The $k$ that controls the vicinity size for $k$-NN exploration is set as 16. We run the Sinkhorn Algorithm [17] for 100 iterations to calculate the overlap-aware Sinkhorn distance and solve the optimal transport for fine matching.

**Coarse Matching** The coarse matching is achieved by calculating the Gaussian correlation matrix [10] as the matching score $\mathbf{S}$. Given the features $\mathbf{F_x}$ and $\mathbf{F_y}$ from the last Geometric Transformer block [10], the matching score is calculated as:

$$\mathbf{S}_{ij} = \exp(-\|\mathbf{F}_{\mathbf{x}_i} - \mathbf{F}_{\mathbf{y}_j}\|_2^2). \tag{18}$$

Then a dual-softmax normalization operator [18, 19, 10] is utilized on $\mathbf{S}$ to reduce ambiguous matches while converting it to the probability $\mathbf{P}$ of mutual matching [19]:

$$\mathbf{P}_{ij} = \text{softmax}(\mathbf{S}(i, \cdot))_j \cdot \text{softmax}(\mathbf{S}(\cdot, j))_i. \tag{19}$$

Finally, we select the top-$k$ largest entries as the coarse correspondence set:

$$\tilde{\mathbf{C}} = \{(\mathbf{x}_i, \mathbf{y}_j) \mid (i, j) \in \text{top-k}_{i,j}(\mathbf{P}_{ij})\}. \tag{20}$$

**Fine Matching** At fine level matching, each correspondence is refined to point level for establishing finer correspondences. For each coarse correspondence $\mathbf{x}_i$ and $\mathbf{y}_j$, we adopt the point-to-node grouping [9, 10] to extract the local patches $\mathbf{X}_i$ and $\mathbf{Y}_j$ from the dense point layer. We then compute the feature similarity matrix $\mathbf{s}$ with their corresponding features $\mathbf{F}_{\mathbf{X}_i}$ and $\mathbf{F}_{\mathbf{Y}_j}$ of two patches:

$$\mathbf{s} = \mathbf{F}_{\mathbf{X}_i}(\mathbf{F}_{\mathbf{Y}_j})^T / \sqrt{d}, \tag{21}$$

where $d$ is the feature dimension.

Then the feature similarity matrix is augmented by a learnable row and column entry that serves as the dustbin as in [20, 9, 10]. The fine matching is considered an optimal transport problem solving by the Sinkhorn Algorithm [21]. We then apply 100 iterations of row- and column-normalization (Sinkhorn Algorithm) to compute the final assigned matching score $\bar{\mathbf{s}}$ by discarding the dustbin entries. The output point-level correspondences from patches $\mathbf{X}_i$ and $\mathbf{Y}_j$ are then obtained by selecting the mutual top-$k$ entries of the score $\bar{\mathbf{s}}$:

$$\mathcal{C} = \{(x_i, y_j) \mid (i, j) \in \text{mutual top-k}_{i,j}(\bar{\mathbf{s}}_{ij})\}. \tag{22}$$

## A.5 Additional Experimental Results

### A.5.1 Scene-wise Results

We present the scene-wise Registration Recall (RR) results in Tab. 8 and Tab. 9. The results on 3DMatch [1] demonstrate our method outperforms baselines in most of the scenes. On the hard case

like Home_2, our method improves the previous best by 3.2 pp, indicating its better registration accuracy. While on 3DLoMatch [8], our method surpasses the strongest baseline [10] in almost all scenarios and has achieved nearly all of the best performances. This indicates that our HST shows even greater performance improvement in low-overlap scenarios compared to high-overlap scenarios, proving its superior robustness in handling severe conditions.

Table 8: Scene-wise registration results on 3DMatch.

| Model | 3DMatch | | | | | | | | |
| | Kitchen | Home_1 | Home_2 | Hotel_1 | Hotel_2 | Hotel_3 | Study | Lab | Mean |
| --- | --- | --- | --- | --- | --- | --- | --- | --- | --- |
| *Registration Recall (%)* ↑ | | | | | | | | | |
| 3DSN [22] | 90.6 | 90.6 | 65.4 | 89.6 | 82.1 | 80.8 | 68.4 | 60.0 | 78.4 |
| FCGF [13] | 98.0 | 94.3 | 68.6 | 96.7 | 91.0 | 84.6 | 76.1 | 71.1 | 85.1 |
| D3Feat [12] | 96.0 | 86.8 | 67.3 | 90.7 | 88.5 | 80.8 | 78.2 | 64.4 | 81.6 |
| Predator [8] | 97.6 | 97.2 | 74.8 | **98.9** | **96.2** | 88.5 | 85.9 | 73.3 | 89.0 |
| CoFiNet [9] | 96.4 | **99.1** | 73.6 | 95.6 | 91.0 | 84.6 | 89.7 | 84.4 | 89.3 |
| GeoTrans [10] | 96.9 | 97.7 | 81.1 | 98.0 | 89.7 | 88.5 | 88.9 | **88.9** | 91.5 |
| HST (**Ours**) | **98.2** | 98.1 | **84.3** | 97.3 | 94.9 | **92.3** | 91.5 | 88.9 | **93.2** |

Table 9: Scene-wise registration results on 3DLoMatch.

| Model | 3DLoMatch | | | | | | | | |
| | Kitchen | Home_1 | Home_2 | Hotel_1 | Hotel_2 | Hotel_3 | Study | Lab | Mean |
| --- | --- | --- | --- | --- | --- | --- | --- | --- | --- |
| *Registration Recall (%)* ↑ | | | | | | | | | |
| 3DSN [22] | 51.4 | 25.9 | 44.1 | 41.1 | 30.7 | 36.6 | 14.0 | 20.3 | 33.0 |
| FCGF [13] | 60.8 | 42.2 | 53.6 | 53.1 | 38.0 | 26.8 | 16.1 | 30.4 | 40.1 |
| D3Feat [12] | 49.7 | 37.2 | 47.3 | 47.8 | 36.5 | 31.7 | 15.7 | 31.9 | 37.2 |
| Predator [8] | 71.5 | 58.2 | 60.8 | 77.5 | 64.2 | 61.0 | 45.8 | 39.1 | 59.8 |
| CoFiNet [9] | 76.7 | 66.7 | 64.0 | 81.3 | 65.0 | 63.4 | 53.4 | 49.6 | 67.5 |
| GeoTrans [10] | 85.9 | 73.5 | 72.5 | 89.5 | 73.2 | **73.2** | **66.7** | **75.7** | 74.0 |
| HST (**Ours**) | **89.1** | **76.7** | **74.8** | **91.0** | **75.4** | 71.4 | 64.1 | **75.7** | **77.3** |

#### A.5.2 Additional Ablation Studies

In this subsection, we provide more ablation studies on the type of Local Exploration, and the type of Patch Feature Transformation. Here, we provide only the Registration Recall results for comparison, as our primary focus is the actual effectiveness of point cloud registration. In many cases, the distribution, location, and number of correspondences can greatly affect the Inlier Ratio (IR) and Feature Matching Recall (FMR) metrics. It is common for the IR and FMR to decrease significantly even with better registration results (higher RR), which can impact the ablative analysis of each component. The similar observations have also been made by [8], [9], and [23].

Table 10: Registration results of additional ablation studies.

| Ablation Part | 3DMatch RR(%) | 3DLoMatch RR(%) |
| --- | --- | --- |
| P2N grouping [9, 10] | 93.2 | 77.2 |
| $k$-NN grouping | **93.5** | **77.8** |
| EdgeConv [24] | **93.5** | 77.5 |
| Self-attention [25] | 93.3 | 77.2 |
| Borderless EdgeConv | **93.5** | **77.8** |

**The type of Local Exploration.** We evaluate two different types of local exploration for extracting local patches: the Point-to-node (P2N in the table) grouping [26, 9], and the $k$-NN grouping we

use. Point-to-node grouping constructs local patches by associating each point with its nearest node (superpoint). It can be easily extended to hierarchical local exploration by replacing nodes with lower spatial resolution point clouds from adjacent layers. The results in Tab. 10 show that the Point-to-node grouping strategy has a slightly lower performance than $k$-NN grouping. In fact, there is not much difference in performance between the two grouping methods. Our experience believe that the two patches extracted by $k$- NN from same source point cloud pair may have closer distributions, which is beneficial for subsequent overlap prediction. Therefore, we choose $k$- NN for local exploration.

**The type of Patch Feature Transformation.** We then provide an ablation study on the patch feature transformation method. We compare the results between the vanilla EdgeConv [24], self-attention [25], and our expanded version of EdgeConv (borderless EdgeConv) in Tab. 10. Registration results show EdgeConv-based methods perform better than self-attention. Due to the introduction of points outside the patch to smooth the feature description and the construction of $k$-NN graph, the borderless one has a slight performance improvement compared to vanilla EdgeConv on 3DLoMatch.

## A.6 Limitations.

The limitations of our proposed HST are 2-fold: 1) HST still involves evaluating and selecting generated coarse correspondences. In some extreme low-overlap or noisy cases, when the quality of putative correspondences searched by the coarse stage model is poor, it can still negatively impact registration and lead to failure. A feasible solution is to subsequently integrate HST into the coarse correspondence generation process to produce a more accurate match set directly. 2) Assessing patch similarity purely from a geometric perspective is limited due to noise, locality, etc. If more priors or features, such as color, normal maps, and semantic labels, could be introduced subsequently to characterize neighborhood features, it would enable a more accurate coarse correspondence searching. We think it is a promising topic for further improving the robustness of registration.

## A.7 Broader Impact.

We study the problem of retrieving accurate correspondence through multi-scale consistency (MSC) for robust point cloud registration and present a novel method to model the MSC called HST. HST makes a first attempt towards injecting a coarse correspondence filter into the coarse-to-fine pipeline, allowing for more accurate and robust point cloud registration. It contributes to various applications such as autonomous driving and robotics. For example, Simultaneous Localization and Mapping (SLAM) tasks can benefit from our method by enabling more robust unified scene reconstruction. Also, our method can help more precise scene understanding in autonomous driving as it is capable of forming a reliable correspondence set for aligning point cloud scans at different timestamps. However potential negative impacts may occur as it is the fundamental of various computer vision tasks. For example, our method may fail in some severe environments like no overlapping area leading to wrong scene representations.

## A.8 Qualitative Results

We provide qualitative results on 3DMatch, 3DLoMatch, and Outdoor KITTI Odometry in Fig. 6. The column (a) and (b) are the input source and target point clouds for registration. Column (c) shows the estimated transformation from our proposed HST while column (d) is the ground truth alignment.

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

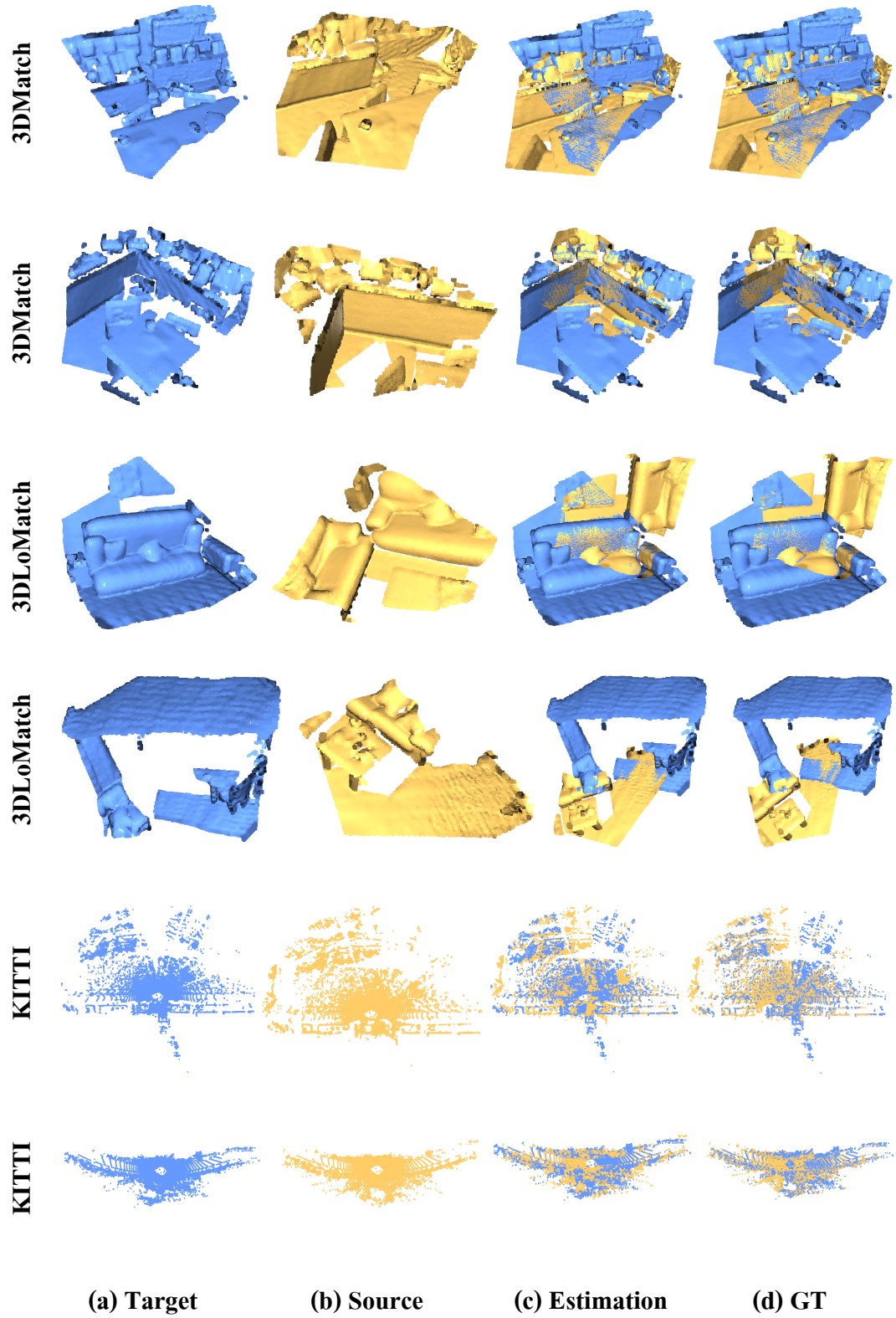

**(a) Target**     **(b) Source**     **(c) Estimation**     **(d) GT**

Figure 6: Qualitative registration results on 3DMatch, 3DLoMatch, and KITTI Odometry.

