# OpenReview forum: "Multi-scale Consistency for Robust 3D Registration via Hierarchical Sinkhorn Tree"
_NeurIPS.cc/2024/Conference — NeurIPS 2024 poster_

### Official Review · Reviewer_hZ2Q · 2024-07-06

**Soundness:** 3
**Presentation:** 3
**Contribution:** 3
**Rating:** 6
**Confidence:** 4

**Summary:**

This paper proposed a hierchical skinhorn tree approach to extract the correspondences that are consistent across multiple feature scales for the point cloud registration task. Besides, an overlap-aware module is proposed to better locate the correspondences around the overlap regions. The proposed methods are evaluated on two benchmarks and its results show superior performance than existing SOTA methods in terms of success rate and registration accuracy.

**Strengths:**

1. It applied the mulit-scale consistency concept to the point cloud registration task. Previous criteria for extreacting correspondences are either based on nearest distance in feature space or geometric consistency in Euclidean space. Here, it extends the former criteria in a multi-scale level. And its experiment results validates its effectiveness.

2. It extends the skinhorn algorithm to the multi scale level to model the multi-scale consistency and proposed a BFS way to optimize the assignment problem in multi-scale levels.

**Weaknesses:**

1. The use of wrong terminology. In Eq1, it seems describe how to perform feature matching across multiple scales, not describing “inlier correspondence”, which is defined as two points of a match are close enough under rigid transformation.

2.  In Eq 5, the variable Tij is not explained. Even though its definition can be found in the Superglue paper, which is a binary assignment variable indicating whether it is a good or bad match, it is never a bad thing to explain more.

3. For sec 3.2.4, it is better to illustrate how to optimize the proposed HST in BFS way with a detailed figure or a piece of pseudo code.

**Questions:**

I don’t have any question.

**Limitations:**

Multi-scale consistency is a quite strong constrains, which will lead to no match or erroneous matches when the learned multi-scale features are less descriptiveness or the test data are extremely low overlap cases. Moreover, the optimization of multi-scale consistency requires a careful fine tuning of hyperparameters, such as #levels, grid size.

---

> ### Author Rebuttal · Authors · 2024-08-06
>
> We thank the reviewer for your valuable comments and suggestions.
>
> **Response to Weakness 1**: Thank you for pointing out this issue. We apologize for any misunderstanding caused by the term "inlier correspondence" in Eq. (1). It should be revised as "putative inlier correspondences". As you mentioned, the definition of inlier correspondence refers to points in two point clouds that are sufficiently close given a ground truth transformation. Actually our original intention in using 'inlier correspondence' here was to convey that, since the ground truth transformation is not provided beforehand, we often have to design a method to estimate the inlier correspondence set. A common approach is to do feature matching, as you mentioned. Therefore, in Eq. (1) we aim to describe how we could solve this problem from the Multi-scale Consistency (MSC) perspective, so the "inlier correspondence" here should be revised as "putative inlier correspondence", representing the inlier correspondence inferred by MSC. We will fix it to "putative inlier correspondence" in the revised version.
>
> **Response to Weakness 2**: Thanks for your feedback regarding the lack of explanation of $T_{ij}$. Here $T_{ij}$ is the element from the assignment matrix $T \in R^{(|\mu_{ov}|+1)\times(|\nu_{ov}|+1)}$ of optimal transport problem, indicating the pairwise matching (overlap) score from source to target point cloud. The row and column sum of the matrix $T$ are respectively subject to the augmented marginal distributions $\mu_{ov}$ and $\nu_{ov}$ to ensure the allocation constraints. $T$ is the optimization variable in the calculation of the Overlap-aware Sinkhorn Distance and can be efficiently solved by the Sinkhorn algorithm. We will include the above explanation in the revised version.
>
> **Response to Weakness 3**: Thank you for your suggestions on better illustrating the optimization of HST. We will include a detailed diagram in the revised version that illustrate the forward computation process and the backward gradient flow to thoroughly demonstrate the optimization process of HST.
>
> **Response to Limitations** : Thank you for your insightful feedback on our work!
> We understand your concern that Multi-scale Consistency (MSC) may hinder the model's ability to learn multi-scale descriptive features because it requires the features of two point clouds to exhibit similarity across multiple scales. In fact, we also observe this issue in our experiments. Therefore, we introduce a layer-wise overlap-aware circle loss (please refer to lines 253-255 in the paper) to help maintain the descriptiveness of features at each layer. This ensures that the backbone does not learn overly uniform features while still preserving MSC.
> As for the concern that MSC might fail in test data with extremely low overlap, we design a set of experiments targeting overlap ratios of less than 10% and compare the performance with GeoTransformer to validate the robustness of HST when facing extremely low overlap. Please note that the currently available preprocessed datasets, 3DMatch (overlap > 30%) and 3DLoMatch (10% < overlap < 30%), do not include samples with such extremely low overlap (overlap<10%). Therefore, we access the 3DMatch raw dataset and collect a set of point cloud pairs with overlap < 10%, which we refer to as 3DExtremeLoMatch (3DExMatch) dataset, containing a total of 1,343 samples. We first test the model pre-trained on 3DMatch directly on 3DExMatch, and the results can be found in the following table:
> |Estimator|Method|RR(\%)|RRE($^\circ$)|RTE(m)|
> |---|---|:---:|:---:|:---:|
> |LGR|GeoTrans|29.8|3.83|0.110|
> ||HST|32.5|3.78|0.111|
> |RANSAC-250|GeoTrans|28.8|4.56|0.123|
> ||HST|31.5|4.42|0.122|
> |RANSAC-500|GeoTrans|30.5|4.25|0.121|
> ||HST|32.5|4.16|0.118|
> |RANSAC-1000|GeoTrans|31.7|4.23|0.118|
> ||HST|33.6|4.03|0.114|
> |RANSAC-2500|GeoTrans|31.9|4.09|0.116|
> ||HST|34.7|3.81|0.111|
> |RANSAC-5000|GeoTrans|31.0|4.04|0.117|
> ||HST|34.4|3.87|0.114|
>
> We then randomly divide the 3DExMatch into training, validation, and test sets with proportions of 60% (805 samples), 10% (134 samples), and 30% (404 samples), respectively. We fine-tune GeoTransformer and HST for 3 epochs on the training set, and then evaluate the models on the test set. The results can be found in the following table. Both results clearly demonstrate that HST maintains strong performance even under low overlap conditions, confirming the effectiveness of our method.
> |Estimator|Method|RR(\%)|RRE($^\circ$)|RTE(m)|
> |---|---|:---:|:---:|:---:|
> |LGR|GeoTrans|53.8|3.73|0.107|
> ||HST|59.0|3.79|0.106|
> |RANSAC-250|GeoTrans|47.0|4.39|0.121|
> ||HST|49.8|4.43|0.119|
> |RANSAC-500|GeoTrans|49.9|4.55|0.113|
> ||HST|52.6|4.03|0.117|
> |RANSAC-1000|GeoTrans|52.4|4.03|0.117|
> ||HST|57.5|4.13|0.115|
> |RANSAC-2500|GeoTrans|55.1|3.89|0.116|
> ||HST|59.0|3.89|0.112|
> |RANSAC-5000|GeoTrans|56.1|4.07|0.116|
> ||HST|60.3|3.93|0.111|
>
> Regarding the potential need to adjust hyperparameters like the number of levels and grid size for optimization, we believe that the adjustment of the number of levels primarily involves balancing performance and latency. As shown in the ablation study in Table 4, increasing the number of levels enhances performance but also slightly increases inference time. If the application scenario requires more real-time processing, the number of levels can be reduced to accelerate the model without significantly sacrificing performance. Similar to other coarse-to-fine methods, such as GeoTransformer, hyperparameters like grid size also require fine-tuning based on the specific scenario and the source of the point clouds.
>
> All the above discussions will be added in the revised version. We hope the above responses address your concerns.

---

> > ### Comment · Reviewer_hZ2Q · 2024-08-12
> >
> > The rebuttal and additional experiments address most of my concerns. I will keep the original score.

---

> > > ### Author Response · Authors · 2024-08-13
> > >
> > > Thank you for acknowledging our rebuttal and the additional experiments we conducted in response to your concerns. We will include the above discussions in the revised version. We appreciate your consideration and the time you took to reassess our work based on these updates.
> > >
> > > Thanks again for your careful review and valuable comments to help us improve our submission.

---

### Official Review · Reviewer_WyrU · 2024-07-07

**Soundness:** 3
**Presentation:** 3
**Contribution:** 2
**Rating:** 5
**Confidence:** 4

**Summary:**

This paper presents a method to enhance the performance of GeoTransformer by filtering outlier correspondences at the coarse level using a multi-scale, overlap-guided Sinkhorn algorithm. It introduces an overlap-aware Sinkhorn Distance designed to detect potential overlapping points, thereby enhancing the robustness of consistency calculations and reducing the complexity of solutions.

**Strengths:**

1. The paper introduces a coarse-level outlier removal method that can be integrated into the GeoTransformer and trained end-to-end with the correspondence search. This approach is intriguing to me, despite being incremental.

2. It proposes a method for modeling multi-scale consistency named HST, which characterizes the similarity of potential overlapping points in neighboring areas, layer by layer, and aggregates these into a multi-scale consistency framework.

3. The numerical results achieve state-of-the-art performance on both indoor and outdoor benchmarks.

**Weaknesses:**

The method appears to be an incremental enhancement, adding a multi-scale local consistency module to the GeoTransformer, which increases both model and time complexity. The registration results still heavily depend on the coarse matching that generates initial correspondences via the GeoTransformer. If these initial correspondences are of poor quality, then performance suffers.

The paper lacks an ablation study on the parameter selection for the top-q predicted scores and the overlap points, which could provide deeper insights into the model's sensitivity to these parameters.

Additionally, it would be beneficial to explore the impact of replacing the Hierarchical Sinkhorn Tree with a direct outlier removal method such as MAC. Understanding the performance differences could highlight the advantages or disadvantages of the proposed method in handling outliers.

The overlap score-guided Sinkhorn algorithm for registration, which was also used in OCFNet, should be cited, even though the application method differs. Additionally, this method functions as an outlier removal technique, similar to approaches like FastMac [2]. Therefore,  many sota outlier remove methods need to be compared.

The authors claim that the proposed method mitigates the effects of low overlap and high noise as its main contribution. However, it appears unable to address the challenges faced by the GeoTransformer in cases of extremely low overlap during registration. It would be beneficial to know how the method performs when the overlap ratio between two point clouds is less than 10%.

[1] Mei, Guofeng, et al. "Overlap-guided coarse-to-fine correspondence prediction for point cloud registration." 2022 IEEE International Conference on Multimedia and Expo (ICME). IEEE, 2022.
[2]YZhang, Yifei, et al. "FastMAC: Stochastic Spectral Sampling of Correspondence Graph." Proceedings of the IEEE/CVF Conference on Computer Vision and Pattern Recognition. 2024.

**Questions:**

1. How about if directly use a outlier removal method such as MAC, FastMac, GCRANSAC to replace the Hierarchical Sinkhorn Tree, what is the performance?
2. The authors claim that the proposed method mitigates the effects of low overlap and high noise as its main contribution. However, it appears unable to address the challenges faced by the GeoTransformer in cases of extremely low overlap during registration. It would be beneficial to know how the method performs when the overlap ratio between two point clouds is less than 10%.
3. Equation (2) appears to be incorrect and requires further explanation.

**Limitations:**

The authors adequately addressed the limitations.

---

> ### Author Rebuttal · Authors · 2024-08-06
>
> We first would like to thank the reviewer for giving us valuable comments.
>
> **Response to the comments on our work (first paragraph of Weaknesses)**: Thank you for your insightful comments on our work. We understand your concern that our method builds upon the foundation laid by existing work. However, we would like to emphasize that modeling the multi-scale consistency (MSC) is non-trivial due to the challenges in ensuring both effectiveness and efficiency. To achieve this, we proposed a series of methods to improve performance while maintaining low complexity, such as a lightweight overlap prediction module, Overlap-aware SD with Overlap Filtering and Overlap-aware Marginals, pruned HST, and others. Our extensive experiments on both indoor and outdoor benchmarks have demonstrated their scene-agnostic effectiveness. HST shows significant improvement compared to SOTAs, maintaining robust performance even under low overlap (please refer to the 3DLoMatch results in Tab. 1 & 2, as well as our response to Question 2 below), while introducing only a slight, acceptable time overhead.
> As for your concern that we might suffer from poor initial correspondences, we believe that by modeling MSC, our method improves the quality of the correspondence set and thus suffers less performance degradation compared to others. All current coarse-to-fine-based methods heavily depend on coarse matching, and if the coarse correspondences are poor, they will face performance loss. Please refer to our response to Weakness 2. Under extremely low overlap, GeoTransformer shows significant performance degradation, whereas HST suffers less.
>
>
> **Response to Question 1**: Thanks for your suggestions about comparing HST with outlier rejection methods. We are also very interested in the comparison. For fairness, we replace HST directly with GC-RANSAC, MAC, and FastMAC. Results on both 3DMatch and 3DLoMatch can be found in the following table.
>
> |3DM\|3DLM|RR(\%)|RRE($^\circ$)|RTE(m)|RR(\%)|RRE($^\circ$)|RTE(m)|time(s)|
> |---|:---:|:---:|:---:|:---:|:---:|:---:|:---:|
> |vanilla|91.5|1.91|0.068|74.0|2.95|0.090|0.260|
> |GC-RANSAC|92.1|1.78|0.068|73.4|2.96|0.088|0.324|
> |MAC|92.2|1.99|0.067|74.4|2.85|0.086|0.475|
> |FastMAC|91.9|1.73|0.062|74.2|2.86|0.087|0.282|
> |HST|93.2|1.70|0.059|77.3|2.71|0.084|0.296|
>
> **Response to Question 2**: Thank you for your constructive suggestions to test the HST under extremely low overlap. We design a set of experiments targeting overlap ratios of less than 10% and compare with GeoTransformer. Please note that the currently available preprocessed datasets, 3DMatch (overlap>30%) and 3DLoMatch (10%<overlap<30%), both do not include samples with such extremely low overlap. Therefore, we access the 3DMatch raw dataset and collect a set of pairs with overlap<10%, which we refer to as 3DExtremeLoMatch (3DExMatch), containing a total of 1,343 samples. We first test the model pre-trained on 3DMatch directly on 3DExMatch, and the results can be found in the following table:
>
> |Estimator|Method|RR(\%)|RRE($^\circ$)|RTE(m)|
> |---|---|:---:|:---:|:---:|
> |LGR|GeoTrans|29.8|3.83|0.110|
> ||HST|32.5|3.78|0.111|
> |RANSAC-250|GeoTrans|28.8|4.56|0.123|
> ||HST|31.5|4.42|0.122|
> |RANSAC-500|GeoTrans|30.5|4.25|0.121|
> ||HST|32.5|4.16|0.118|
> |RANSAC-1000|GeoTrans|31.7|4.23|0.118|
> ||HST|33.6|4.03|0.114|
> |RANSAC-2500|GeoTrans|31.9|4.09|0.116|
> ||HST|34.7|3.81|0.111|
> |RANSAC-5000|GeoTrans|31.0|4.04|0.117|
> ||HST|34.4|3.87|0.114|
>
> We then randomly divide the 3DExMatch into training, validation, and test sets with proportions of 60% (805 samples), 10% (134 samples), and 30% (404 samples), respectively. We fine-tune GeoTransformer and HST both for 3 epochs on the training set and then evaluate on the test set. The results are as follows:
>
> |Estimator|Method|RR(\%)|RRE($^\circ$)|RTE(m)|
> |---|---|:---:|:---:|:---:|
> |LGR|GeoTrans|53.8|3.73|0.107|
> ||HST|59.0|3.79|0.106|
> |RANSAC-250|GeoTrans|47.0|4.39|0.121|
> ||HST|49.8|4.43|0.119|
> |RANSAC-500|GeoTrans|49.9|4.55|0.113|
> ||HST|52.6|4.03|0.117|
> |RANSAC-1000|GeoTrans|52.4|4.03|0.117|
> ||HST|57.5|4.13|0.115|
> |RANSAC-2500|GeoTrans|55.1|3.89|0.116|
> ||HST|59.0|3.89|0.112|
> |RANSAC-5000|GeoTrans|56.1|4.07|0.116|
> ||HST|60.3|3.93|0.111|
>
> **Response to Question 3**: Thank you for your feedback on Eq. (2). We apologize for the typo and for any misunderstandings it may have caused.
> Our purpose of Eq. (2) is to find the k-NN of any given point $x_i^{l}$ in its next layer to form the local patch. We provide the following explanations in hopes of resolving your confusion.
> (1) We mistakenly used $argmax$ instead of $argmin$ operator in the formula. We should use the $argmin$ on $K$ to find the k nearest points to form the local patch.
> (2) Our initial intention is to use a more mathematical $argmin$ operator to represent the k-NN search instead of directly using $kNN(\cdot)$. However, such an expression might be hard to understand and could lead to ambiguities, such as the definition of the set $S$ in the formula. Therefore, we take a compromise approach by replacing the $argmin$ with the $argtopk$ operator to make it more intuitive.
> The following is the revised Eq. (2) and will be included in the revised version.
> Given $\mathbf{x}_i^{(l)}$, its local patch $\mathbf{P}_i^{(l+1)}$ is defined as:
> $$
> \mathbf{P}_i^{(l+1)}=\operatorname{argtopk} _{\mathbf{x}_j^{(l+1)}\in\mathbf{X}^{(l+1)}}(-||\mathbf{x}_i^{(l)},\mathbf{x}_j^{(l+1)}||_2)
> $$
>
> **Lack of ablation on top-q**: We apologize for the lack of ablation on top-q for overlap points filtering. The performance of HST is not sensitive to the selection of q. The RR of varying q can be found in the following table.
> |Top-q|5|10|15|25|30|35|40|45|50|
> |---|:---:|:---:|:---:|:---:|:---:|:---:|:---:|:---:|:---:|
> |3DMatch|91.2|92.4|92.6|92.9|92.5|92.7|93.1|92.5|91.5|91.8|
> |3DLoMatch|75.4|76.5|76.8|77.2|77.6|77.1|77.3|77.3|76.8|76.4|
>
> **Citing other overlap score-guided Sinkhorn method**: Thank you for your suggestion. We will cite OCFNet in our revised version.

---

> > ### Comment · Reviewer_WyrU · 2024-08-13
> > **Post-Rebuttal**
> >
> > I appreciate the authors' clarifications.  My concerns have been addressed.

---

> > > ### Author Response · Authors · 2024-08-13
> > >
> > > Thank you for confirming that your concerns have been addressed. Your insightful comments have been instrumental in enhancing the clarity and quality of our work.
> > >
> > > Thank you once again for your constructive feedback and positive evaluation.

---

### Official Review · Reviewer_Sd2B · 2024-07-11

**Soundness:** 2
**Presentation:** 3
**Contribution:** 2
**Rating:** 5
**Confidence:** 4

**Summary:**

This paper studies the problem of correspondence retrieval for point cloud registration. To this end, this paper proposes the Hierarchical Sinkhorn Tree, which is a pruned tree structure designed to hierarchically measure the local consistency of each coarse correspondences. To validate the proposed methods, the authors conducted experiments on both indoor and outdoor benchmarks.

**Strengths:**

1. This paper is well-written, well-organized, and easy to follow;

2. The idea of introducing tree structure into a coarse-to-fine mechanism is somehow interesting.

**Weaknesses:**

1. Although incorporating a tree structure into a coarse-to-fine mechanism is interesting, the idea sounds more like repeating the coarse-to-fine matching strategy used by CoFiNet and GeoTransformer. I suggest emphasizing this core contribution in the rebuttal phase;

2. In the main comparison (Tab. 1), the proposed method does not significantly outperform existing methods, which limits the value of the model.

3. As the proposed model leverages a tree structure to repeat the coarse-to-fine procedure, the efficiency (in terms of running speed) is a major concern. The authors should include related comparisons with state-of-the-art approaches.

**Questions:**

See the weaknesses part.

**Limitations:**

Limitations have been discussed in the Appendix.

---

> ### Author Rebuttal · Authors · 2024-08-06
>
> We thank the reviewer for the insightful comments and feedback. We hope that our responses can address your concerns.
>
> **Response to Weakness 1**: Thanks you for your valuable comments. First, we would like to emphasize that modeling multi-scale consistency (MSC) is non-trivial, even with the help of a coarse-to-fine strategy for modeling the neighborhood. We understand your concern that our method might simply be applying the coarse-to-fine procedure across multiple scales. But our proposed method is not simply a repetition of the coarse-to-fine strategy, as merely repeating such an operation would result in an $O(n^l)$ complexity, where $l$ is the number of scales. This level of complexity is unacceptable, posing a major challenge in practical applications. Additionally, this naive approach could introduce a large number of non-overlapping noisy points, negatively impacting the model's robustness and performance, and could even degrade the quality of the correspondence set compared to the original.
> To address these issues, we proposed a series of designs to improve the model's feasibility and performance, which distinguishes it from the mentioned models. First, we design a lightweight Patch Overlap Prediction module to efficiently predict overlap. Next, we propose Overlap-aware Sinkhorn Distance with Overlap Points Filtering and Overlap-aware Marginal Prior to effectively retain the most likely overlapping points based on the prediction, filtering out a large number of potential non-overlapping points. Finally, we perform the above methods with local exploration layer-wisely to prune the K-ary tree rooted at the superpoint, thereby constructing the HST to model the MSC of correspondences. Our extensive experiments on both indoor and outdoor benchmarks have demonstrated the scene-agnostic effectiveness of our proposed method.
> Finally, we would like to emphasize our key contributions as follows: 1) To the best of our belief, our work is the first to introduce MSC into point cloud registration tasks to mitigate the effects of low overlap and high noise. 2) We propose a method for modeling MSC called HST, which introduces a pruned tree structure to efficiently characterize the similarity of potential overlapping points in the vicinity areas layer by layer and aggregates them into MSC. 3) We introduce an overlap-aware Sinkhorn Distance with Overlap Points Filtering and Overlap-aware Marginal Prior to focus optimal transport processes only on potential overlapping points, significantly enhancing the robustness of consistency calculations while reducing solution complexity.
> The above discussions will be included in the revised version.
>
> **Response to Weakness 2**: Thank you for your comments regarding the comparative performance. As mentioned by other reviewers and shown in Tab 1, HST gains significant improvements over the state-of-the-art methods on both indoor and outdoor benchmarks. The improvements are even greater on 3DLoMatch, where point cloud pairs share fewer overlaps, suggesting that our model demonstrates better robustness and performance under low-overlap, high-noise scenarios. To validate this, we further design a set of experiments targeting overlap ratios of less than 10% and compare the performance with GeoTransformer. Please note that the currently available preprocessed datasets, 3DMatch (overlap > 30%) and 3DLoMatch (10% < overlap < 30%), do not include samples with such extremely low overlap (overlap<10%). Therefore, we access the 3DMatch raw dataset and collect a set of point cloud pairs with overlap < 10%, which we refer to as 3DExtremeLoMatch (3DExMatch) dataset, containing a total of 1,343 samples. We first test the model pre-trained on 3DMatch directly on 3DExMatch, and the results can be found in the following table:
> |Estimator|Method|RR(\%)|RRE($^\circ$)|RTE(m)|
> |---|---|:---:|:---:|:---:|
> |LGR|GeoTrans|29.8|3.83|0.110|
> ||HST|32.5|3.78|0.111|
> |RANSAC-250|GeoTrans|28.8|4.56|0.123|
> ||HST|31.5|4.42|0.122|
> |RANSAC-500|GeoTrans|30.5|4.25|0.121|
> ||HST|32.5|4.16|0.118|
> |RANSAC-1000|GeoTrans|31.7|4.23|0.118|
> ||HST|33.6|4.03|0.114|
> |RANSAC-2500|GeoTrans|31.9|4.09|0.116|
> ||HST|34.7|3.81|0.111|
> |RANSAC-5000|GeoTrans|31.0|4.04|0.117|
> ||HST|34.4|3.87|0.114|
>
> We then randomly divide the 3DExMatch into training, validation, and test sets with proportions of 60% (805 samples), 10% (134 samples), and 30% (404 samples), respectively. We fine-tune GeoTransformer and HST both for 3 epochs on the training set, and then evaluate the models on the test set. The results can be found in the following table. Both results clearly demonstrate that HST can significantly outperform other methods even under low overlap conditions, further confirming the effectiveness of our method.
> |Estimator|Method|RR(\%)|RRE($^\circ$)|RTE(m)|
> |---|---|:---:|:---:|:---:|
> |LGR|GeoTrans|53.8|3.73|0.107|
> ||HST|59.0|3.79|0.106|
> |RANSAC-250|GeoTrans|47.0|4.39|0.121|
> ||HST|49.8|4.43|0.119|
> |RANSAC-500|GeoTrans|49.9|4.55|0.113|
> ||HST|52.6|4.03|0.117|
> |RANSAC-1000|GeoTrans|52.4|4.03|0.117|
> ||HST|57.5|4.13|0.115|
> |RANSAC-2500|GeoTrans|55.1|3.89|0.116|
> ||HST|59.0|3.89|0.112|
> |RANSAC-5000|GeoTrans|56.1|4.07|0.116|
> ||HST|60.3|3.93|0.111|
>
> **Response to Weakness 3**: Thank you for your insightful feedback regarding the efficiency of our proposed method. We share your concern on this matter. In fact, we have compared the running time of our method with state-of-the-art approaches under different estimators in the manuscript. Please refer to the last column of Table 2. The results indicate that our method achieves significant performance improvements with only a slight, acceptable increase in latency due to the introduction of new modules. We will highlight this comparison more prominently in the revised version.
>
> Thank you again for your valuable feedback. We hope our responses address your concerns.

---

> > ### Comment · Reviewer_Sd2B · 2024-08-13
> >
> > Thanks the authors for the response. I think at this moment, most of my concerns have been addressed. I am leaning towards the positive side.

---

> > > ### Author Response · Authors · 2024-08-13
> > >
> > > Thank you for your encouraging feedback and for acknowledging the revisions and responses we have provided. We are glad to hear that our explanations have addressed your concerns and would like to extend our sincere thanks for the upgrade in your evaluation score. We will add those revisions to the revised version.
> > >
> > > Thank you once again for your thoughtful and constructive review.

---

### Official Review · Reviewer_AXAW · 2024-07-13

**Soundness:** 3
**Presentation:** 3
**Contribution:** 3
**Rating:** 6
**Confidence:** 4

**Summary:**

This paper introduces the Hierarchical Sinkhorn Tree (HST) for reliable correspondence identification in point cloud registration. The core idea is to hierarchically evaluate the local consistency of each correspondence at multiple feature scales using Sinkhorn distance, thereby filtering out the locally dissimilar correspondences. Specifically, Local Exploration is first employed to extract local patches at each correspondence’s next decoder layer. Then, the overlap-aware Sinkhorn distance is used to evaluate the patch differences, filtering out the non-overlapping local patches. The filtered overlapping points are further utilized for additional local exploration and overlap Sinkhorn distance measurements. Finally, the consistency measures across all scales are aggregated for robust transformation estimation. Extensive experiments on public benchmark datasets verify the effectiveness of the proposed method.

**Strengths:**

(1) The authors’ introduction of the concept of multi-scale consistency (MSC) for robust registration is both interesting and promising. Developing a Hierarchical Sinkhorn Tree to model MSC is also a novel approach that achieves significant performance gains.

(2) Overall, this manuscript is well-structured and clearly written. The authors have effectively organized their ideas, making the content easy to follow and understand.

(3) The authors provide sufficient comparisons and ablation studies to verify the effectiveness of the proposed mechanism.

**Weaknesses:**

(1) Could you please explain Equation (2)? I understand that you aim to gather the k-nearest neighbor points from the next layer to form the local patch  X_i^{(l)} . However, I find it difficult to grasp this idea from Equation (2). For instance, what does  \|x_i^l, x_j^{l+1}\|  represent? Do you mean the distance between  x_i^l  and  x_j^{l+1} ? If so, how does applying the argmin operator on K yield the desired local patch? It’s confusing and not intuitive.

(2) Some relevant outlier rejection-focused registration methods should be cited, such as:
[1] Robust Outlier Rejection for 3D Registration with Variational Bayes. CVPR’2023

**Questions:**

See Weaknesses.

**Limitations:**

See Weaknesses.

---

> ### Author Rebuttal · Authors · 2024-08-06
>
> We first would like to thank the reviewer for providing us valuable comments and suggestions.
>
> **Response to Weakness 1**: Thank you for your question regarding Eq. (2) about the k-NN local exploration part. We apologize for the typo in Eq. (2) and for any misunderstandings it may have caused.
> Your are correct about the understanding of Eq. (2). Its purpose is to search for the k-NN of any given point $x_i^{l}$ from the $l$-th layer in the next layer to form the corresponding local patch. We provide the following explanations in hopes of resolving your confusion.
> (1) The term $||x_i^l, x_j^{l+1} ||$ represents the Euclidean distance between the given $l$-th layer's point $x_i^l$ and the $(l+1)$-th layer's point $x_j^{l+1}$. So the operator $||\cdot||$ we used here is actually the L2-norm of coordinates' difference. We will add a subscript 2 to the $||\cdot||$ operator to avoid any potential misunderstandings, i.e., revise $|| x_i^l, x_j^{l+1} ||$ to $|| x_i^l, x_j^{l+1} ||_2$.
> (2) We mistakenly used $argmax$ instead of $argmin$ operator in the formula. The set $K$ actually represents the collection of Euclidean distances between all points in $(l+1)$-th layer and the given point in $l$-th layer. We should use the $argmin$ operator on $K$ to search for the k nearest points to form the local patch.
> (3) Our initial intention is to use a more mathematical $argmin$ operator to represent the k-NN search instead of directly using $kNN(\cdot)$. However, such an expression might be difficult to understand and could lead to ambiguities, such as the definition of the set $S$ in the formula. Therefore, we take a compromise approach by replacing the $argmin$ operator with the $argtopk$ operator to make it more intuitive and easier to understand.
> The following is the revised formula and will be included in the revised version.
>
> Given $\mathbf{x}_i^{(l)}$, its local patch $\mathbf{P}_i^{(l+1)}$ is defined as:
> $$
> \mathbf{P}_i^{(l+1)}=\operatorname{argtopk} _{\mathbf{x}_j^{(l+1)}\in\mathbf{X}^{(l+1)}}(-||\mathbf{x}_i^{(l)},\mathbf{x}_j^{(l+1)}||_2).
> $$
>
> **Response to Weakness 2**: Thanks for your suggestion about citing relevant outlier rejection-focused registration methods. We will cite approaches like GC-RANSAC [1], MAC [2], VBReg [3], FastMAC [4], etc., and summarize them as fine-level outlier removal methods in our revised version.
> [1] Barath, Daniel, and Jiří Matas. "Graph-cut RANSAC." Proceedings of the IEEE conference on computer vision and pattern recognition. 2018.
> [2] Zhang, Xiyu, et al. "3D registration with maximal cliques." Proceedings of the IEEE/CVF Conference on Computer Vision and Pattern Recognition. 2023.
> [3] Jiang, Haobo, et al. "Robust outlier rejection for 3d registration with variational bayes." Proceedings of the IEEE/CVF conference on computer vision and pattern recognition. 2023.
> [4] Zhang, Yifei, et al. "FastMAC: Stochastic Spectral Sampling of Correspondence Graph." Proceedings of the IEEE/CVF Conference on Computer Vision and Pattern Recognition. 2024.

---

### Author Rebuttal · Authors · 2024-08-06

We first would like to thank all the reviewers for providing insightful comments and we are immensely grateful for your thorough feedback on our manuscript. It is encouraging that the reviews found

* Our paper is well-written
  - "The authors have effectively organized their ideas, making the content easy to follow and understand." - Reviewer AXAW
  - "This paper is well-written, well-organized, and easy to follow" - Reviewer Sd2B

* Our paper presents a novel and interesting idea
  - "The authors’ introduction of the concept of multi-scale consistency (MSC) for robust registration is both interesting and promising. Developing a Hierarchical Sinkhorn Tree to model MSC is also a novel approach that achieves significant performance gains." - Reviewer AXAW
  - "The idea of introducing tree structure into a coarse-to-fine mechanism is somehow interesting." - Reviewer Sd2B
  - "This approach is intriguing to me" - Reviewer WyrU

* Experiments show superior performance and verify the effectiveness of the proposed method
  - "The authors provide sufficient comparisons and ablation studies to verify the effectiveness of the proposed mechanism." - Reviewer AXAW
  - "The numerical results achieve state-of-the-art performance on both indoor and outdoor benchmarks." - Reviewer - WyrU
  - "The proposed methods are evaluated on two benchmarks and its results show superior performance than existing SOTA methods in terms of success rate and registration accuracy." - Reviewer hZ2Q

---

We have carefully read all the comments and responded to them in detail. All of those will be addressed in the final version.
We summarize the main concerns of the reviews with the corresponding response as follows.

1. **About the performance under extremely low overlap.**
To test the robustness of HST when facing extremely low overlap, we design a set of experiments targeting overlap ratios of less than 10% and compare the performance with GeoTransformer using different estimators. However, we found that currently available preprocessed datasets, 3DMatch (overlap > 30%) and 3DLoMatch (10% < overlap < 30%), do not include samples with such extremely low overlap (overlap<10%). Consequently, we accessed the raw 3DMatch dataset and collected a new set of point cloud pairs with overlap < 10%, which we refer to as the 3DExtremeLoMatch (3DExMatch) dataset, comprising a total of 1,343 samples. We conducted two types of experiments: one where we directly evaluated the pre-trained model and another where we evaluated the model after fine-tuning on the partitioned 3DExMatch dataset. The empirical results demonstrate that our proposed HST outperforms GeoTransformer in both settings, particularly after fine-tuning. This indicates that, even under extremely low overlap conditions, our method does not fail due to a lack of matches or excessive mismatches. Instead, by incorporating MSC modeling, HST can form a higher-quality correspondence set, resulting in improved performance under severe conditions and less performance degradation compared to other methods.

2. **About the comparison with outlier rejection-based methods.**
We have conducted experiments comparing HST with outlier rejection-based methods to gain more insights into handling outliers. For a fair comparison, we replace HST directly with the state-of-the-art approaches GC-RANSAC, MAC, and FastMAC, and evaluate them on both 3DMatch and 3DLoMatch datasets. All three methods, along with HST, showed performance improvements compared to the vanilla GeoTransformer, with HST demonstrating the most significant enhancement. This suggests that when the quality of the correspondence set is high, the registration process benefits from outlier rejection methods. However, on 3DLoMatch, the improvements from these three methods were less pronounced, and GC-RANSAC even showed a potential negative impact. In contrast, HST maintained better robustness, highlighting its superior effectiveness over previous outlier rejection methods in scenarios with low overlap and high noise. This further validates the efficacy of our MSC modeling in handling outliers from coarse correspondences.

Thanks again for your efforts in the review. We appreciate all the valuable feedback that helped us to improve our submission.

---

### Comment · Area_Chair_VPBd · 2024-08-10
**Please respond to the authors' rebuttal.**

Dear Reviewers,

The authors have posted their rebuttals to the reviews. Could you please respond to the rebuttals?
Please engage in the discussion with the authors. Your help is much appreciated.

Thanks,

AC

---

### Decision · Program_Chairs · 2024-09-25

**Decision:**

Accept (poster)

**Comment:**

The paper is about introducing multi-scale consistency for point cloud registration. This is an under-explored problem due to the characteristics of point clouds (i.e., the format being irregular). This paper deals with this problem effectively with the newly proposed hierarchical Sinkhorn tree (HST). Most reviewers have agreed on the importance of the topic, as well as the novelty and good performance of the proposed method. There were some concerns (e.g., incremental contribution, marginal improvements, and processing time), but they were all clearly resolved by the authors' responses. The AC has personally reviewed the paper in detail and agreed on the above points.